# Full reconstruction of simplicial complexes from binary contagion and Ising data

Huan Wang[1,5], Chuang Ma [2,5], Han-Shuang Chen[3], Ying-Cheng Lai [4] & Hai-Feng Zhang [1✉]

Previous efforts on data-based reconstruction focused on complex networks with pairwise or two-body interactions. There is a growing interest in networks with higher-order or many-body interactions, raising the need to reconstruct such networks based on observational data. We develop a general framework combining statistical inference and expectation maximization to fully reconstruct 2-simplicial complexes with two- and three-body interactions based on binary time-series data from two types of discrete-state dynamics. We further articulate a two-step scheme to improve the reconstruction accuracy while significantly reducing the computational load. Through synthetic and real-world 2-simplicial complexes, we validate the framework by showing that all the connections can be faithfully identified and the full topology of the 2-simplicial complexes can be inferred. The effects of noisy data or stochastic disturbance are studied, demonstrating the robustness of the proposed framework.

[1] The Key Laboratory of Intelligent Computing and Signal Processing of Ministry of Education, School of Mathematical Science, Anhui University, Hefei 230601, China. [2] School of Internet, Anhui University, Hefei 230601, China. [3] School of Physics and Material Science, Anhui University, Hefei 230601, China. [4] School of Electrical, Computer and Energy Engineering, Arizona State University, Tempe, AZ 85287, USA. [5] These authors contributed equally: Huan Wang, Chuang Ma. ✉email: haifengzhang1978@gmail.com

In network science and engineering, a subfield of research is to find the network topology and nodal dynamical equations from data[1]. This is important because networks are ubiquitous in the real world but the details of their connection topology and the intrinsic dynamical systems governing the properties and physical observables of the network are often unknown. The details are desired not only for understanding but also for protecting, disabling, or controlling the network dynamical behaviors (depending on the specific applications), and a viable way is to solve the inverse problem of determining the network details through observational data if they are available. As for any inverse problems in mathematics and physical sciences, the network inverse problem is challenging. Previous works in this area focused on "conventional" networks with pairwise interactions only[1–16]. Existing methods include those which are based on drive-response[3,5], adaptive synchronization[2,11], noise correlation[6,15], compressive sensing[7,9,17], maximum likelihood estimation[13,14,16], and Granger causality[4,8]. The data can be from continuous- or discrete-time dynamical processes. For example, the drive-response and adaptive synchronization methods use data from continuous-time nonlinear coupled systems[2,3,5,11], while the maximum likelihood estimation method is suitable for data from discrete-time dynamics[13,14,16]. In this paper, motivated by the fact that higher-order networks have become a state-of-the-art subfield of research in network science[18–24], we develop a reconstruction framework for finding from time-series data network topology with higher-order interactions.

While pairwise or node-to-node interactions are the familiar type in networks, it has been recognized that higher-order interactions are also ubiquitous and important. For example, in a social network, the collective recommendation of multiple friends can often be more persuasive than the recommendation of a single friend to convince the individual to buy a new product. In a rumor spreading process, a piece of false news is likely to be accepted by an individual if it is shared or promoted simultaneously by many people[25–27]. A similar situation occurs in neuronal networks, where a firing event is often the result of excitatory and inhibitory interactions among many neurons. In all these cases, the interaction arises simultaneously among a group of nodes in the network, and to describe the network by the conventional pairwise interactions is no longer adequate[28]: higher-order interactions beyond the pairwise relationship must be taken into account. Mathematically, higher-order interactions can be described as hypergraphs or simplicial complexes[29], i.e., networks containing higher-order simplexes. In particular, a $k$-simplex describes the simultaneous interaction among $(k + 1)$ nodes, where a zero-simplex specifies an isolated node (i.e., without any interaction), a 1-simplex represents the conventional pairwise interaction, a 2-simplex underlies the simultaneous interaction among three nodes, and so on.

The past three years have witnessed a growing interest in higher-order networks[30]. For example, random walks on hypergraphs were studied, where a walker chooses the next destination depending on the number and the size of the shared hyperedges[31]. A family of random walks on hypergraphs with a parameter controlling the bias of the dynamics towards hyperedges of small or large size was constructed and the impacts of walk strategy and walk time on community detection were elucidated[32]. The stability conditions of the general dynamical processes on hypergraphs were found[18], and a social contagion model on hypergraphs was constructed which presents dynamical phenomena such as first- and second-order transitions, bistability and hysteresis[33]. A simplicial model of social contagion was proposed and it was demonstrated that the reinforcement mechanisms in 2-simplex can lead to a discontinuous phase transition[34]. The impacts of the heterogeneity of simplicial complexes on the SIS (susceptible-infected-susceptible) spreading

model with collective and individual contagion were analyzed[35], and a pair approximation theory to study the SIS dynamics in simplicial complexes was developed, which was argued to be more accurate than the Markov-chain and mean field methods[36]. A social communication model including idea integration and information transmission in simplicial complexes was proposed and the critical condition leading to the outbreak of information was identified[37]. A simplicial activity driven model was proposed and the impact of both simplicial and temporally evolving interactions were analyzed[38]. In terms of network reconstruction, a statistical method to detect higher-order interactions from network data of pairwise links has recently been developed[21].

In this paper, we develop a framework to reconstruct complex networks with higher-order interactions from data. To be concrete, we focus on networks with 2-simplexes and assume that the dynamical processes on the network are social contagion and simplicial Ising dynamics that generate binary time-series data. Our method is of the statistical inference type pivoted on maximum likelihood estimation, with the aim to fully reconstruct both pairwise interactions (links) and 2-simplexes at the same time, thereby distinguishing our work from the recent method based on link data[21]. In particular, the central task is to estimate the probabilities of each node connecting to the reconstruction or target node (pairwise interaction) and of any two nodes forming a three-body 2-simplex with the target node. We articulate a two-step process to greatly enhance the computational efficiency and an effective truncation process to determine the final reconstructed structure of the simplicial complex. Using three synthetic and four real-world simplicial complexes, we demonstrate the accuracy of our reconstruction method and establish its robustness with respect to variations in the average degree of the network and stochastic fluctuations. Our work represents an initial effort in reconstructing complex networks with higher-order interactions based on observed time-series data.

## Results

**Simplicial complexes.** A $k$-simplex $\sigma$ is formed by a filled clique of a set of $k + 1$ nodes $[v_0, \cdots, v_k]$, which defines a $(k + 1)$-body interaction[39]. A 1-simplex is two nodes connected by an edge, a 2-simplex is three nodes connected pairwisely by edges and with an additional single face, i.e., a triangle, and a 3-simplex is four vertices connected pairwisely by edges and joined by four faces, which are filled in to form a solid tetrahedron, and so on. A simplicial complex $\mathcal{K}$ composed of a set of nodes $\mathcal{V}$ is a collection of simplexes, with the additional requirement[39,40] that if a simplex is in $\mathcal{K}$ ($\sigma \in \mathcal{K}$), then any simplex $\varrho$ composed of subsets of simplex $\sigma$ should also be included in $\mathcal{K}$. For example, a 2-simplicial complex $\mathcal{K}$ is a collection of 0-, 1- and 2-simplexes.

**Social contagion dynamics.** Peer influence and reinforcement mechanisms are ubiquitous in the dynamical process of social contagion[41], from which higher-order interactions in the network are originated. A social contagion model taking reinforcement into account on 2-simplicial complexes was proposed[34], which exploits the SIS type of spreading dynamics with binary-state dynamical variables. In particular, let $S_i^t$ be the state of node $i$ at time $t$. Each node has two possible states: susceptible ($S_i^t = 0$) or infected ($S_i^t = 1$). At the initial time, a fraction $\rho_0$ of nodes is infected. A susceptible node $i$ can get infection from an infected neighbor $j$ through their pairwise interaction $(i, j)$ with probability $\beta_1$. Node $i$ can also be infected through a 2-simplex $(i, j, k)$, where both $j$ and $k$ have already been infected, with the probability $\beta_2$, and this event can be understood as a synergistic reinforcement effect. For convenience, we set $\beta_1 = \alpha/k_1$ and $\beta_2 = \omega/k_2$, where $\alpha$ and $\omega$ are two non-zero positive constants, $k_1$ and $k_2$ are the

average degrees of the two-body and three-body interactions in a 2-simplicial complex, respectively. In general, we have $\alpha < \omega$ so as to ensure that the role of 2-simplex is embodied in the spreading dynamics. Each infected node recovers to the susceptible state with the probability $\mu$. In our work, the values of $\beta_1$ and $\beta_2$ are selected near their respective thresholds to facilitate efficient and accurate reconstruction of 2-simplicial complexes. The effects of varying the values of $\beta_1$ and $\beta_2$ on the reconstruction accuracy are also studied (Sec. I of Supplementary Information (SI)).

**Simplicial Ising dynamics**. The Ising model arises in many fields due to its fundamental role in phase transitions in statistical physics. It has also been applied to many social systems[42,43]. While the Ising dynamics on networks have been extensively studied[44–46], previous studies were exclusively conducted for networks with pairwise interactions only. To our knowledge, Ising dynamics on networks with higher-order interactions have not been studied.

To address the synergistic reinforcement effect of 2-simplex, we define a simplicial Ising dynamics on 2-simplicial complexes. Each node has two possible states: spin-down ($S_i^t = -1$) or spin-up ($S_i^t = +1$). At the initial time, the state of each node $i$ is randomly assigned as $+1$ or $-1$ with equal probability. Defining the Hamiltonian as:

$$H(t) = -J_1 \sum_{(i,j)} S_i^t S_j^t - J_2 \sum_{(i,j,k)} S_i^t S_j^t S_k^t, \qquad (1)$$

where $J_1$ and $J_2$ are the strengths of two-body and three-body interactions, and $(i,j)$ and $(i,j,k)$ denote the two-body and the three-body connections in the 2-simplicial complex, respectively. The first term in the Hamiltonian characterizes the interaction between the edges (i.e., two-body connections) and the second term contains three-body interactions from the 2-simplex. At each time step, the spin-flipping probability of each node $i$ is given by $(1 + e^{\delta \Delta E_i^t})^{-1}$, where $\delta$ is the inverse temperature. The quantity

$$\Delta E_i^t = 2J_1 \sum_{(i,j) \in \partial_i} S_i^t S_j^t + 2J_2 \sum_{(i,j,k) \in \nabla_i} S_i^t S_j^t S_k^t$$

represents the change in the energy caused by a flipping of node $i$ at time $t$, where $\partial_i$ and $\nabla_i$ are the 1-simplex set and the 2-simplex set containing node $i$, respectively.

**Statistical inference framework**. For SIS and Ising processes taking place on a 2-simplicial complex of size $N$, the available time-series data representing the states of nodes at different time steps can be stored in a data matrix $S$, where each row is a time string representing all nodes' states at that time step and each column is a node's state at different time steps. We reconstruct 2-simplicial complexes from the data matrix $S$ with our statistical inference framework. This task consists of three steps: (1) establishing the likelihood function based on the available data matrix $S$; (2) obtaining the connection probabilities of two- and three-body interactions by maximizing the likelihood function according to the idea of the expectation maximization (EM) method, and (3) executing an improved two-step reconstruction strategy to significantly increase the computational efficiency. The details of the framework are described in Methods.

**Quantification of reconstruction performance**. We use F1 score to quantify the reconstruction accuracy[47], a global performance indicator defined as

$$\text{F1} = \frac{2P * R}{P + R}, \qquad (2)$$

where $P = \text{TP}/(\text{TP} + \text{FP})$ and $R = \text{TP}/(\text{TP} + \text{FN})$, with TP, FP, TN, FN being the numbers of true positive, false positive, true negative and false negative classes, respectively. A larger value of F1

corresponds to a higher accuracy and F1 = 1 indicates that the original network structure has been fully reconstructed with zero error.

**Reconstructing synthetic and real-world simplicial complexes**. For readability, the results from the social contagion dynamics are presented in the main text, while those from the simplicial Ising dynamics are presented in Sec. III of SI.

Figure 1 presents results of reconstructing three synthetic 2-simplicial complexes (see Sec. "Construction of synthetic and real-world 2-simplicial complexes" in Methods on how these networks are constructed), where squares, diamonds and circles denote the performance of reconstructing the two-body connections while triangles with different orientations demonstrate the performance of reconstructing three-body connections. Several features can be seen from Fig. 1. First, the reconstruction accuracy increases with the length $T$ of the time series and can reach the unity value for $T \gtrsim 8000$. Second, the average degrees $k_1$ and $k_2$ of two-body and three-body simplexes, respectively, have different effects on the reconstruction accuracy. In particular, as shown in Fig. 1a–c, a small value of $k_1$ tends to increase the reconstruction accuracy of both types of simplexes. This can be understood by noting that a small value of $k_1$ means that there are fewer two-body connections that need to be reconstructed, thereby enhancing the accuracy of the two-body connections for the same length of the time series. At the same time, fewer two-body connections reduce the complexity in reconstructing three-body connections and thereby improving the reconstruction accuracy. Regarding the effects of $k_2$, Fig. 1d–f reveal that its value affects only the reconstruction accuracy of three-body connections and has little effect on the accuracy of reconstructing two-body connections that have no dependence on the three-body connections in a 2-simplicial complex. Third, the reconstruction accuracy of three-body interactions is lower than that of two-body interactions owing to the complicated structure of former and its dependence on the latter.

Figure 2 shows the results of reconstructing four real-world 2-simplicial complexes: Hypertext2009[48], Thiers12[49], InVS15[50], and LyonSchool[51,52] (see Sec. "Construction of synthetic and real-world 2-simplicial complexes" in Methods for the details of these real-world networks). The basic parameters of these 2-simplicial complexes constructed from the datasets are listed in Table 1. It can be seen from Fig. 2 that, as for the real-world networks, the reconstruction accuracies for both the two-body and three-body interactions increase with the length of the time series. Remarkably, these network structures are quite irregular, complicating the reconstruction. Nonetheless, for $T = 20,000$, the F1 score can exceed 80%.

An issue of practical significance is the robustness of our reconstruction framework against random perturbations. To address this issue, we randomly flip a fraction $f$ of infected states and the same number of susceptible states in the data matrix $S$ (see Sec. "Details of the statistical inference framework" in Methods) and investigate the effect of $f$ on the reconstruction accuracy as characterized by F1. The results are shown in Fig. 3 for three synthetic 2-simplicial complexes and three real-world 2-simplicial complexes. It can be seen that increasing the fraction $f$ of flipping leads to a reduction of F1. In particular, the value of F1 for the two-body connections can be as high as 50% even when 30% of the infected states have been flipped ($f = 0.3$), attesting to the robustness of our framework in reconstructing pairwise links against stochastic fluctuations in the data.

**Discussion**

To find the network structure from observational data has been an active research field for more than fifteen years[1]. In previous

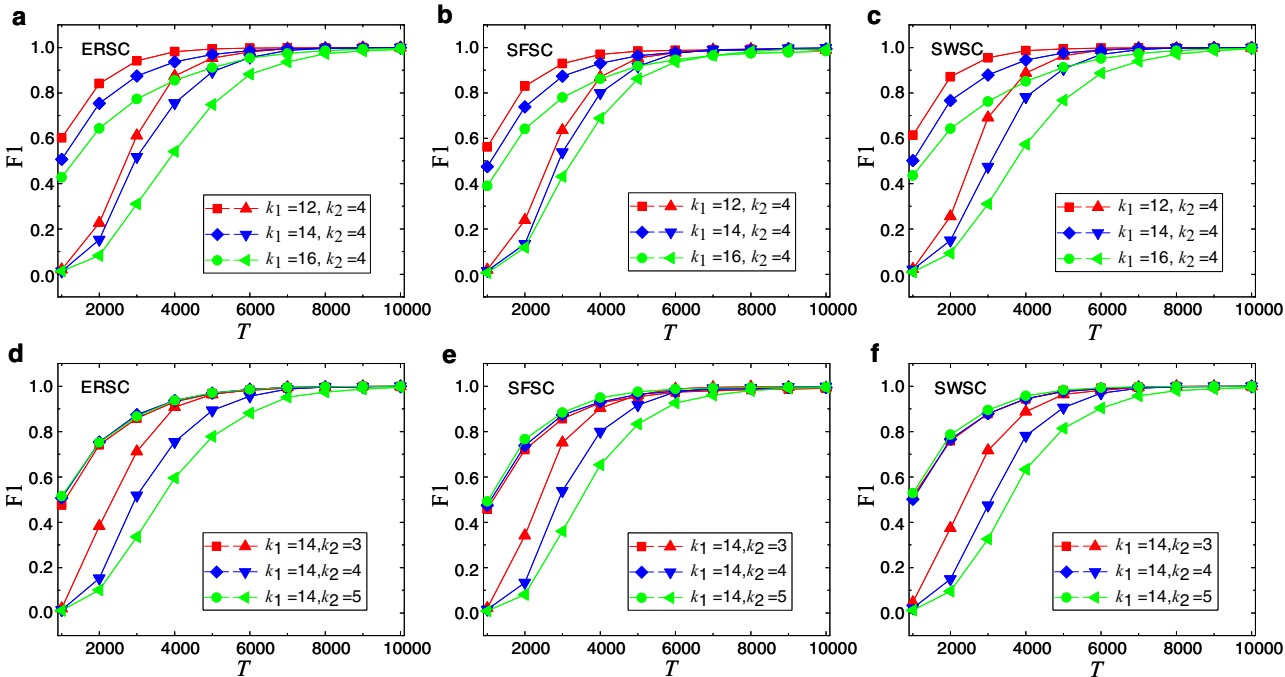

**Fig. 1 Reconstruction performance for synthetic 2-simplicial complexes.** Shown is F1 score as a function of the length $T$ of the observational binary time series for three synthetic 2-simplicial complexes: (**a**, **d**) random simplicial complex (ERSC), (**b**, **e**) scale-free simplicial complex (SFSC), and (**c**, **f**) small-world simplicial complex (SWSC). In each panel, squares, diamonds and circles denote the performance of reconstructing the two-body connections while triangles with different orientations demonstrate the performance of reconstructing three-body connections, and different values of the average degree are distinguished by colors. All simplicial complexes have the same size $N = 200$. Other parameter values are $\alpha = 0.8$, $\omega = 2.4$, $\rho_0 = 0.2$, and $\mu = 1$. The results are averaged over five realizations.

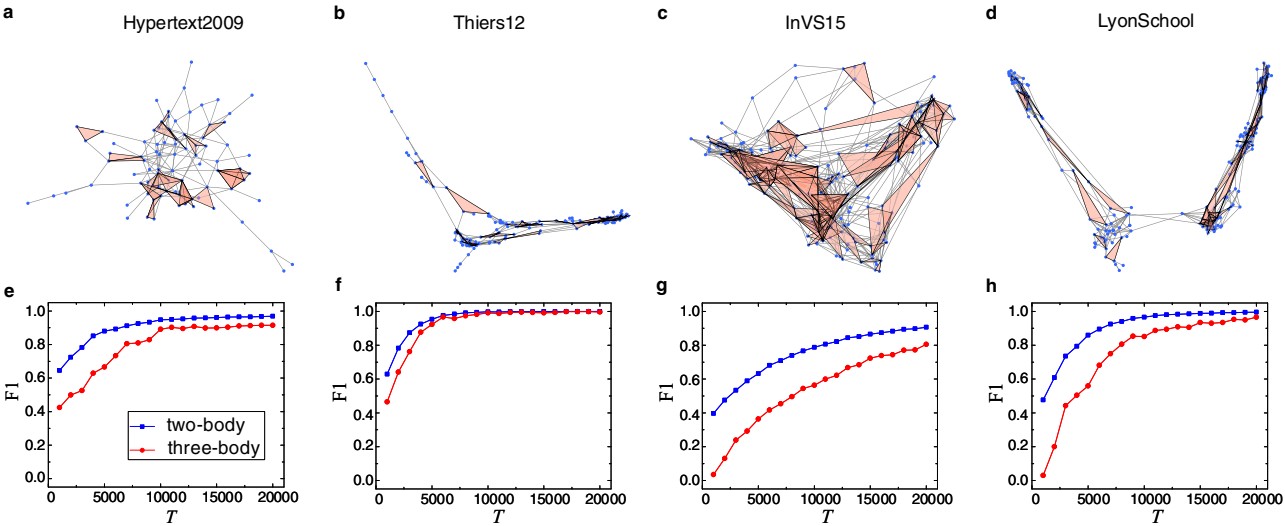

**Fig. 2 Visualization and reconstruction performance for real-world 2-simplicial complexes.** Visualization results for (**a**) Hypertext2009, (**b**) Thiers12, (**c**) InVS15, and (**d**) LyonSchool. The corresponding reconstruction performances are shown in panels (**e**–**h**) in terms of F1 score versus the length $T$ of the available time series. The blue squares and red circles demonstrate the reconstruction performance of two-body and three-body connections, respectively. Parameter values are $\alpha = 0.3$, $\omega = 1$, $\rho_0 = 0.2$, and $\mu = 1$. Each data point is the result of averaging over five realizations.

studies, the term "network structure" is largely referred to as the collection of pairwise connections as characterized by the adjacency matrix of the network. Since the goal is to figure out whether there is a link between any two nodes, the existing methods focused on measures that are suitable for ascertaining the "two-body" interactions, such as those based on pairwise correlation or synchronization. From the beginning of modern network science and engineering slightly over two decades ago,

networks with only pairwise connections have represented the standard setting of study. Likewise, the inverse problem of data-based discovery of the network structure has been exclusively carried out in this setting. To our knowledge, in the current literature, the problem of finding higher-order connections in complex networks from time-series data has not been addressed.

Higher-order interactions are nonetheless ubiquitous in complex networks and its importance has been gradually recognized

with an accumulating interest, eventually generating an explosive growth of research recently[18–24]. The structure of networks with higher-order connections, also known as simplicial complexes, are represented by tensors of high orders. For example, three-body interactions or 2-simplexes in a network can be described by a tensor of rank 3. Structurally, simplicial complexes are significantly more sophisticated than the conventional networks with pairwise links only, and richer dynamics can be anticipated in the former, which have begun to be studied. From the point of view of inverse problem, to reconstruct simplicial complexes from time-series data represents a great challenge.

We have taken the first step to address this inverse problem. Focusing on complex networks with 2-simplexes, we have developed a statistical inference framework by which all two-body and three-body interactions in the network can be found simultaneously from binary time-series data only, i.e., no prior

knowledge about the network to be reconstructed is required. The backbone of our reconstruction framework is maximum likelihood estimation that yields the probabilities of all the possible pairwise and three-body connections and a criterion to associate the probabilities with the actual interactions. To significantly increase the computational efficiency, we have proposed and tested a two-step process and a truncation process to determine the true structure of the simplicial complexes. The reconstruction framework has withstood tests on synthetic and real-world simplicial complexes with respect to accuracy and robustness against random fluctuations.

Many open problems remain. For example, our reconstruction framework is formulated in terms of binary time-series data from social contagion dynamics and simplicial Ising dynamics (Sec. III of SI). How to reconstruct higher-order networks from data generated by different dynamical processes needs to be studied. Also, our statistical inference method is developed for 2-simplicial complexes that are perhaps the "simplest" network structure beyond the conventional networks with pairwise interactions. To reconstruct networks with higher-order interactions such as 3-simplicial complexes and hypergraphs is worth pursuing. It is also necessary to develop methods to improve the reconstruction accuracy with shorter time series. We hope our work will stimulate further research in this emerging subfield of data-based reconstruction of complex networks with higher-order interactions.

## Methods

**Details of the statistical inference framework**. We describe the details of our statistical inference framework through an illustrative example, as shown in Fig. 4, where a 2-simplicial complex with $N = 30$ nodes and its data matrix are illustrated

**Table 1 Basic parameters of the four 2-simplicial complexes constructed from real-world datasets.**

| Data set | Context | $N$ | $k_1$ | $k_2$ | $\zeta$ |
|---|---|---|---|---|---|
| Hypertext2009 | Conference | 85 | 4.52 | 1.16 | 20 |
| Thiers12 | High school | 156 | 4.56 | 1.21 | 20 |
| InVS15 | Workplace | 211 | 7.52 | 2.19 | 20 |
| LyonSchool | Primary school | 222 | 5.42 | 2.18 | 50 |

$N$ is the number of nodes, $k_1$ and $k_2$ are the average degrees of two-body and three-body connections, respectively, $\zeta$ is a threshold to filter out certain connections with low interaction frequency.

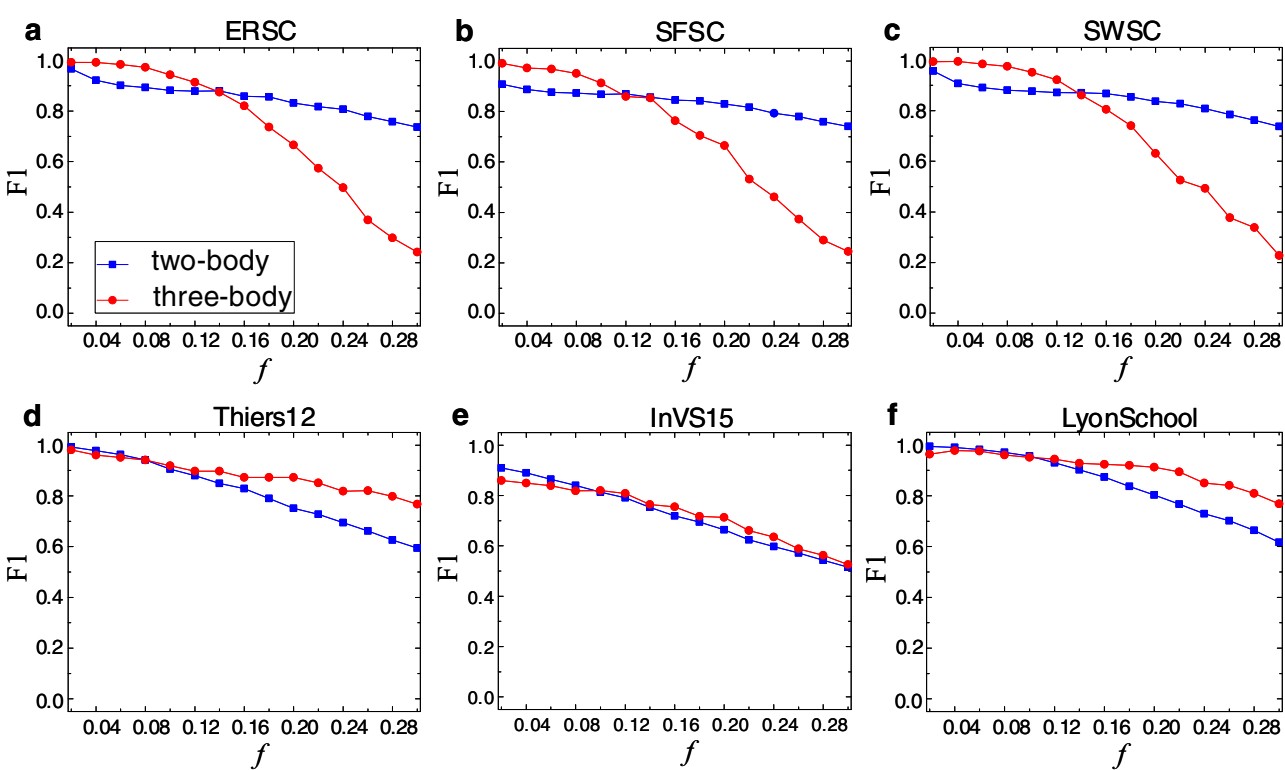

**Fig. 3 Effect of random flipping ratio $f$ on reconstruction performance for synthetic and real-world simplicial complexes.** Shown is the F1 score for (**a**) random simplicial complex (ERSC), (**b**) scale-free simplicial complex (SFSC), (**c**) small-world simplicial complex (SWSC), (**d**) Thiers12, (**e**) InVS15, and (**f**) LyonSchool. The blue squares and red circles demonstrate the reconstruction performance of two-body and three-body connections, respectively. The parameter values for the synthetic simplicial complexes are $N = 200$, $k_1 = 12$, $k_2 = 4$, $T = 10,000$, $\alpha = 0.8$, $\omega = 2.4$, $\rho_0 = 0.2$, and $\mu = 1$. For the real-world simplicial complexes, the parameter values are $T = 20,000$, $\alpha = 0.3$, $\omega = 1$, $\rho_0 = 0.2$, and $\mu = 1$. Each data point is the result of averaging over ten realizations.

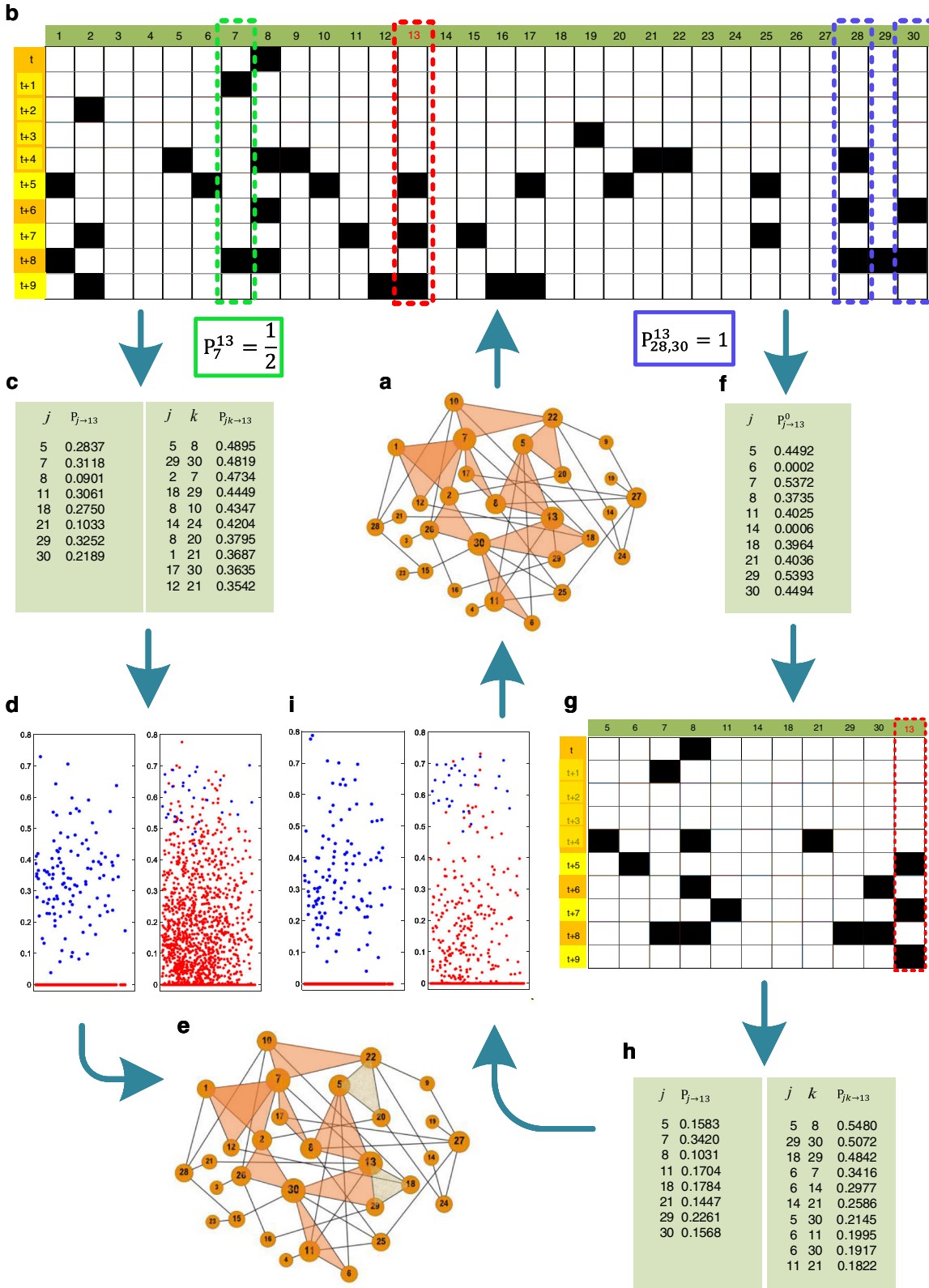

in Fig. 4a, b, respectively. For such a network hosting SIS dynamics, the probability of a susceptible node $i$ (i.e., $S_i^t = 0$) being infected (i.e., $S_i^{t+1} = 1$) is determined only by the infected neighbors and the infected 2-simplexes in which two other nodes in the 2-simplex are both infected, at time $t$. The transition probability from the infected state to the susceptible state does not depend on the states of the neighbors, so it is only necessary to consider the transition probability from the susceptible state to the infected state for constructing the network. We stress that the details of the dynamical process, such as the infection probabilities $\beta_1$ and $\beta_2$ as

well as the recovery probability $\mu$, are assumed to be unknown but only the binary time series of the nodal states are available. Figure 4 presents an illustrative example to describe the details of our method.

*Establishing the likelihood function.* Let $j \rightarrow i$ denote the event that node $j$ has a direct impact on the state of node $i$. For example, node $j$ can directly spread the virus or send a piece of information to node $i$, which means that node $j$ is one of immediate neighbors of node $i$. Nodes $i$ and $j$ thus form a 1-simplex, a property

**Fig. 4 Schematic illustration of reconstructing a 2-simplicial complex based on binary social contagion. a** A 2-simplicial complex of size $N = 30$, where the black links represent 1-simplexes and the orange shadows represent 2-simplexes. **b** Data matrix $S$ that stores all nodes' states at different time steps, where each row is a time string representing all nodes' states at that time step and each column is a node's state at different time steps, where the black and blank squares denote the 1 and 0 states, respectively. Take node 13 as an example (the red frame), the values of $P_j^{13}(\forall j \neq 13)$ and $P_{jk}^{13}(\forall j \neq k \neq 13)$ can be calculated from data matrix $S$: $P_7^{13} = 1/2$ (the green frame) and $P_{28,30}^{13} = 1$ (the purple frame). **c** The values of $P_{j \to 13}$ and $P_{jk \to 13}$ are obtained through the EM algorithm, where only non-zero values of $P_{j \to 13}$ and the top 10 values of $P_{jk \to 13}$ are shown. **d** The values of $P_{j \to i}$ (each column in the left subgraph) and $P_{jk \to i}$ (each column in the right subgraph) for each node $i$ based on the method described in Secs. 4.1.1 and 4.1.2, where the blue and red dots denote the actual and nonexistent two-body or three-body connections, respectively. **e** The 2-simplicial complex is inferred based on the probabilities in (**d**), in which the two-body connections are exactly predicted, but two 2-simplexes (5, 20, 22) and (13, 18, 29) in (**a**) cannot be predicted (marked by the light-yellow shadows). **f** The values of $P_{j \to 13}^0$ for node 13 obtained by iterating Eqs. 22–25, and only non-zero values are shown. **g** Compressed data matrix that records only the columns in $S$ giving $P_{j \to 13}^0 > 0$. **h** The values of $P_{j \to 13}$ and $P_{jk \to 13}$ for node 13 based on the compressed data in (**g**). **i** The values of $P_{j \to i}$ and $P_{jk \to i}$ for each node $i$. Finally, the full 2-simplicial complex in (**a**) can be exactly reconstructed by determining whether $P_{j \to i} > 0$ or $P_{jk \to i} > \hat{\Delta}_i$.

---

that is independent of time $t$. Similarly, let $jk \to i$ denote the event that the synergistic reinforcement effect coming from nodes $j$ and $k$ has a direct impact on the state of node $i$, which is also independent of time. In the following, we determine the probabilities of node $i$ and node $j$ being connected and of three nodes $i, j, k$ forming a three-body connection $(i, j, k)$.

The conditional probability of $S_i^{t+1} = 1$ and $j \to i$ given $S_j^t = 1$ and $S_i^t = 0$ can be written as

$$
\begin{aligned}
& P\left(S_i^{t+1} = 1, j \to i \middle| S_i^t = 0, S_j^t = 1\right) \\
& = P\left(j \to i \middle| S_i^t = 0, S_j^t = 1, S_i^{t+1} = 1\right) * P\left(S_i^{t+1} = 1 \middle| S_i^t = 0, S_j^t = 1\right) \\
& = P_{j \to i} P_j^i,
\end{aligned}
\tag{3}
$$

where $P_{j \to i} \triangleq P(j \to i | S_i^t = 0, S_j^t = 1, S_i^{t+1} = 1)$ is the probability of node $i$ being infected by node $j$ under the conditions $S_i^t = 0$, $S_j^t = 1$ and $S_i^{t+1} = 1$, $P_{j \to i} > 0$ indicates that node $j$ is a neighbor of node $i$, and $P_j^i \triangleq P(S_i^{t+1} = 1 | S_i^t = 0, S_j^t = 1)$ is the probability of $S_i^{t+1} = 1$ under the conditions $S_i^t = 0$ and $S_j^t = 1$, which can be estimated from the data matrix $S$. Take the matrix in Fig. 4b as an example and suppose we wish to estimate the value of $P_7^{13}$, where nodes 13 (i.e., node 13 is the target node) and 7 are highlighted by red and green frames, respectively. It is necessary to extract each pair including the time string with $S_{13}^t = 0$ and its next time strings at $t + 1$. It can be seen that seven pairs of time strings can be extracted: $(t, t+1), (t+1, t+2), (t+2, t+3), (t+3, t+4), (t+4, t+5), (t+6, t+7)$, and $(t+8, t+9)$. It can also be seen that two time moments: $t+1$ and $t+8$, satisfy the conditions that node 13 is in the susceptible state and node 7 is in the infected state. The only time at which node 13 can be infected at the next time step is $t+8$. As a result, we have $P_7^{13} = 1/2$.

Similarly, the conditional probability of $S_i^{t+1} = 1$ and $jk \to i$ given $S_j^t S_k^t = 1$ and $S_i^t = 0$ can be written as

$$
\begin{aligned}
& P\left(S_i^{t+1} = 1, jk \to i \middle| S_i^t = 0, S_j^t S_k^t = 1\right) \\
& = P\left(jk \to i \middle| S_i^t = 0, S_j^t S_k^t = 1, S_i^{t+1} = 1\right) * P\left(S_i^{t+1} = 1 \middle| S_i^t = 0, S_j^t S_k^t = 1\right) \\
& = P_{jk \to i} P_{jk}^i,
\end{aligned}
\tag{4}
$$

where $P_{jk \to i} \triangleq P(jk \to i | S_i^t = 0, S_j^t S_k^t = 1, S_i^{t+1} = 1)$ is the probability of node $i$ being infected through the synergistic interaction from nodes $j$ and $k$, under the conditions $S_i^t = 0$, $S_j^t S_k^t = 1$, and $S_i^{t+1} = 1$, and $P_{jk \to i} > 0$ indicates that the three nodes $i, j, k$ form a 2-simplex. The probability $P_{jk}^i \triangleq (S_i^{t+1} = 1 | S_i^t = 0, S_j^t S_k^t = 1)$ can be estimated from the data matrix $S$ in a similar way. Again, take the three nodes 13, 28, and 30 in Fig. 4b as an example. It can be seen that the time instants at which $S_{13}^t = 0$, $S_{28}^t = 1$ and $S_{30}^t = 1$ are fulfilled are $t+6$ and $t+8$. Because $S_{13}^{t+7} = 1$ and $S_{13}^{t+9} = 1$, we have $P_{28,30}^{13} = 1$.

According to Eqs. 3, 4, the expected number of susceptible node $i$ being infected at $t_m + 1$ is given by

$$
\begin{aligned}
E_i^{t_m+1} = & \sum_{j(j \neq i)} P\left(S_i^{t_m+1} = 1, j \to i \middle| S_i^{t_m} = 0, S_j^{t_m} = 1\right) \Psi_j^{t_m} \\
& + \sum_{j,k(j \neq k \neq i)} P\left(S_i^{t_m+1} = 1, jk \to i \middle| S_i^{t_m} = 0, S_j^{t_m} S_k^{t_m} = 1\right) \Psi_{jk}^{t_m} + \varepsilon_i \\
= & \sum_{j(j \neq i)} P_{j \to i} P_j^i \Psi_j^{t_m} + \sum_{j,k(j \neq k \neq i)} P_{jk \to i} P_{jk}^i \Psi_{jk}^{t_m} + \varepsilon_i,
\end{aligned}
\tag{5}
$$

where $\Psi_j^{t_m}$ represents the events that nodes $j$ are infected at time $t_m$, similarly, $\Psi_{jk}^{t_m}$ is the events that both nodes $j$ and $k$ are infected at time $t_m$, and their values are zero or one. For example, if $\Psi_j^{t_m} = 1$, it means that node $j$ is infected at time $t_m$;

otherwise, $\Psi_j^{t_m} = 0$ when it is not infected at time $t_m$. The quantity $\varepsilon_i$ represents the noise due to the errors from the collected data.

In general, the probability of a given number of events occurring in a fixed interval of time is characterized by the Poisson distribution, so we use it to capture the random nature of the times that node $i$ is infected. An advantage of the Poisson distribution is that it makes a mathematical analysis and computations with the EM algorithm feasible[53–56]. Specifically, the likelihood function can be described as

$$
P\left(\left\{\Psi_i^{t_m+1}\right\}_{m=1,\cdots,M} \middle| \Theta, \left\{\Psi_j^{t_m}\right\}_{m=1,\cdots,M;j=1,\cdots,N}\right) = \prod_{m(\Psi_i^{t_m}=0)} \frac{e^{-E_i^{t_m+1}}\left(E_i^{t_m+1}\right)^{\Psi_i^{t_m+1}}}{\Psi_i^{t_m+1}!},
\tag{6}
$$

where $\Theta$ denotes the set of variables $P_{j \to i}$, $P_{jk \to i}$ and $\varepsilon_i$. We have $\Psi_i^{t_m+1}! \equiv 1$ since $\Psi_i^{t_m+1}$ is either zero or one.

*Maximizing the likelihood function based on EM algorithm.* We next use the EM method to maximize the likelihood function[57] for determining the parameter $\Theta$ in Eq. 6. Taking the logarithm form of Eq. 6, we get

$$
L(\Theta) = \sum_{m(\Psi_i^{t_m}=0)} \left(\Psi_i^{t_m+1} \log E_i^{t_m+1} - E_i^{t_m+1}\right) = \sum_{m(\Psi_i^{t_m}=0)} \begin{bmatrix} \Psi_i^{t_m+1} \log\left(\sum_{j(j \neq i)} P_{j \to i} P_j^i \Psi_j^{t_m} \right. \\ \left. + \sum_{j,k(j \neq k \neq i)} P_{jk \to i} P_{jk}^i \Psi_{jk}^{t_m} + \varepsilon_i\right) \\ -\left(\sum_{j(j \neq i)} P_{j \to i} P_j^i \Psi_j^{t_m}\right) \\ + \sum_{j,k(j \neq k \neq i)} P_{jk \to i} P_{jk}^i \Psi_{jk}^{t_m} + \varepsilon_i \end{bmatrix}.
\tag{7}
$$

Applying the Jensen's inequality to the logarithmic term on the right side of Eq. 7 yields

$$
\begin{aligned}
& \log\left(\sum_{j(j \neq i)} P_{j \to i} P_j^i \Psi_j^{t_m} + \sum_{j,k(j \neq k \neq i)} P_{jk \to i} P_{jk}^i \Psi_{jk}^{t_m} + \varepsilon_i\right) \\
& = \log\left(\sum_{j(j \neq i)} \rho_j^{t_m} \frac{P_{j \to i} P_j^i \Psi_j^{t_m}}{\rho_j^{t_m}} + \sum_{j,k(j \neq k \neq i)} \rho_{jk}^{t_m} \frac{P_{jk \to i} P_{jk}^i \Psi_{jk}^{t_m}}{\rho_{jk}^{t_m}} + \rho_{\varepsilon_i}^{t_m} \frac{\varepsilon_i}{\rho_{\varepsilon_i}^{t_m}}\right) \\
& \geq \sum_{j(j \neq i)} \rho_j^{t_m} \log \frac{P_{j \to i} P_j^i \Psi_j^{t_m}}{\rho_j^{t_m}} + \sum_{j,k(j \neq k \neq i)} \rho_{jk}^{t_m} \log \frac{P_{jk \to i} P_{jk}^i \Psi_{jk}^{t_m}}{\rho_{jk}^{t_m}} + \rho_{\varepsilon_i}^{t_m} \log \frac{\varepsilon_i}{\rho_{\varepsilon_i}^{t_m}} \\
& = \sum_{j(j \neq i)} \rho_j^{t_m} \log\left(P_{j \to i} P_j^i \Psi_j^{t_m}\right) + \sum_{j,k(j \neq k \neq i)} \rho_{jk}^{t_m} \log\left(P_{jk \to i} P_{jk}^i \Psi_{jk}^{t_m}\right) \\
& \quad + \rho_{\varepsilon_i}^{t_m} \log \varepsilon_i - \sum_{j(j \neq i)} \rho_j^{t_m} \log \rho_j^{t_m} - \sum_{j,k(j \neq k \neq i)} \rho_{jk}^{t_m} \log \rho_{jk}^{t_m} - \rho_{\varepsilon_i}^{t_m} \log \rho_{\varepsilon_i}^{t_m},
\end{aligned}
\tag{8}
$$

where the equality holds if

$$
\rho_j^{t_m} = \frac{P_{j \to i} P_j^i \Psi_j^{t_m}}{\varepsilon_i + \left(\sum_{j'(j' \neq i)} P_{j' \to i} P_{j'}^i \Psi_{j'}^{t_m} + \sum_{j',k'(j' \neq k' \neq i)} P_{j'k' \to i} P_{j'k'}^i \Psi_{j'k'}^{t_m}\right)},
\tag{9}
$$

$$
\rho_{jk}^{t_m} = \frac{P_{jk \to i} P_{jk}^i \Psi_{jk}^{t_m}}{\varepsilon_i + \left(\sum_{j'(j' \neq i)} P_{j' \to i} P_{j'}^i \Psi_{j'}^{t_m} + \sum_{j',k'(j' \neq k' \neq i)} P_{j'k' \to i} P_{j'k'}^i \Psi_{j'k'}^{t_m}\right)},
\tag{10}
$$

and

$$\rho_{\varepsilon_i}^{t_m} = \frac{\varepsilon_i}{\varepsilon_i + \left( \begin{array}{c} \sum\limits_{j'(j' \neq i)} P_{j' \to i} P_{j'}^i \Psi_{j'}^{t_m} + \\ \sum\limits_{j',k'(j' \neq k' \neq i)} P_{j'k' \to i} P_{j'k'}^i \Psi_{j'k'}^{t_m} \end{array} \right)}. \tag{11}$$

The maximization problem of Eq. 7 can then be transformed into maximizing the following equation:

$$\hat{L}(\Theta) = \sum_{m(\Psi_i^{t_m} = 0)} \sum_{j(j \neq i)} \left( \Psi_i^{t_m+1} \rho_j^{t_m} \log\left( P_{j \to i} P_j^i \Psi_j^{t_m} \right) - \Psi_i^{t_m+1} \rho_j^{t_m} \log \rho_j^{t_m} - P_{j \to i} P_j^i \Psi_j^{t_m} \right)$$
$$+ \sum_{m(\Psi_i^{t_m} = 0)} \sum_{j,k(j \neq k \neq i)} \left( \Psi_i^{t_m+1} \rho_{jk}^{t_m} \log\left( P_{jk \to i} P_{jk}^i \Psi_{jk}^{t_m} \right) - \Psi_i^{t_m+1} \rho_{jk}^{t_m} \log \rho_{jk}^{t_m} - P_{jk \to i} P_{jk}^i \Psi_{jk}^{t_m} \right)$$
$$+ \sum_{m(\Psi_i^{t_m} = 0)} \left( \Psi_i^{t_m+1} \rho_{\varepsilon_i}^{t_m} \log \varepsilon_i - \Psi_i^{t_m+1} \rho_{\varepsilon_i}^{t_m} \log \rho_{\varepsilon_i}^{t_m} - \varepsilon_i \right). \tag{12}$$

Calculating the partial derivative of $\hat{L}(\Theta)$ with respect to $P_{j \to i}$, $P_{jk \to i}$ and $\varepsilon_i$ and setting them to zero, we get

$$\frac{\partial \hat{L}(\Theta)}{\partial P_{j \to i}} = \sum_{m(\Psi_i^{t_m} = 0)} \left( \frac{\Psi_i^{t_m+1} \rho_j^{t_m}}{P_{j \to i}} - P_j^i \Psi_j^{t_m} \right) = 0, \tag{13}$$

$$\frac{\partial \hat{L}(\Theta)}{\partial P_{jk \to i}} = \sum_{m(\Psi_i^{t_m} = 0)} \left( \frac{\Psi_i^{t_m+1} \rho_{jk}^{t_m}}{P_{jk \to i}} - P_{jk}^i \Psi_{jk}^{t_m} \right) = 0, \tag{14}$$

$$\frac{\partial \hat{L}(\Theta)}{\partial \varepsilon_i} = \sum_{m(\Psi_i^{t_m} = 0)} \left( \frac{\Psi_i^{t_m+1} \rho_{\varepsilon_i}^{t_m}}{\varepsilon_i} - 1 \right) = 0, \tag{15}$$

which give

$$P_{j \to i} = \frac{\sum\limits_{m(\Psi_i^{t_m} = 0)} \left( \Psi_i^{t_m+1} \rho_j^{t_m} \right)}{\sum\limits_{m(\Psi_i^{t_m} = 0)} \left( P_j^i \Psi_j^{t_m} \right)}, \tag{16}$$

$$P_{jk \to i} = \frac{\sum\limits_{m(\Psi_i^{t_m} = 0)} \left( \Psi_i^{t_m+1} \rho_{jk}^{t_m} \right)}{\sum\limits_{m(\Psi_i^{t_m} = 0)} \left( P_{jk}^i \Psi_{jk}^{t_m} \right)}, \tag{17}$$

$$\varepsilon_i = \frac{\sum\limits_{m(\Psi_i^{t_m} = 0)} \left( \Psi_i^{t_m+1} \rho_{\varepsilon_i}^{t_m} \right)}{\sum\limits_{m(\Psi_i^{t_m} = 0)} (1)}. \tag{18}$$

The six equations Eqs. 9–11 and Eqs. 16–18 can be used to solve $P_{j \to i}$, $P_{jk \to i}$ and $\varepsilon_i$. In particular, by initializing all values of $P_{j \to i}, P_{jk \to i}, \varepsilon_i (\forall j \neq k \neq i)$ to be one and then calculating the values of $\rho_j^{t_m}, \rho_{jk}^{t_m}$ and $\rho_{\varepsilon_i}^{t_m}$ in Eqs. 9–11, we substitute them into Eqs. 16–18 to find the values of $P_{j \to i}$, $P_{jk \to i}$ and $\varepsilon_i$. We repeat this process until convergence is achieved. Since a single iterative process does not ensure global optimization, we carry out the above iteration process a number of times and choose the proper values that give the maximum of the quantity in Eq. 12.

As an example, as shown in Fig. 4c, the values of $P_{j \to 13}$ and $P_{jk \to 13}$ are given according to this iteration process, where $P_{j \to 13} > 0$ and the top 10 values of $P_{jk \to 13}$ are demonstrated. Similarly, all the values of $P_{j \to i}$ and $P_{jk \to i}$ can be calculated for each node $i$. As presented in Fig. 4d, each column above the abscissa corresponds to the predicted 1-simplex probabilities (the left subgraph of Fig. 4d) and 2-simplex probabilities (the right subgraph of Fig. 4d) of a node, and the blue and red dots denote the actual and nonexistent two-body or three-body connections, respectively.

*An improved two-step reconstruction strategy.* For a 2-simplicial complex structure with $N$ nodes, when predicting the 2-simplexes of a node $i$, we randomly choose two nodes (e.g., $j$ and $k$) and calculate the probability $P_{jk \to i}$, which requires calculating $(0.0ptN-12)$ values. To reduce the computational load and increase the reconstruction accuracy, we articulate an improved two-step strategy. The particularity of simplicial complexes stipulates that the other two nodes forming a 2-simplex with node $i$ must be the neighbors of node $i$, so it is not necessary to calculate the probability $P_{jk \to i}$ if node $j$ or node $k$ is not a neighbor of node $i$. The reconstruction process can then be divided into two steps. At the first step, the "approximate" neighborhood of each node is predicted and their corresponding columns in the data matrix $S$ are extracted, leading to a compressed data matrix. At the second step, based on the compressed data matrix, the values of $P_{j \to i}$ and $P_{jk \to i}$ for each node $i$ are predicted by iterating Eqs. 9–11 and 16–18. Our two-step method was not designed for the general challenging task of consistently inferring all the subfaces for arbitrarily higher-order simplices. In fact, our method requires the closure condition of simplicial complexes: it is necessary to know in advance that the network under reconstruction is a 2-simplicial complex. Given this premise, the two-step strategy infers first the two-body and then the three-body interactions (i.e., 2-simplex) from the inferred two-body interactions. While the two-step method is efficient to reconstruct 2-simplicial complexes, at the present it

cannot be used to reconstruct the hypergraphs because its second step is to find the triangles from the neighbors (i.e., edges).

For the first step, the predicted neighbors are not accurate because the three-body interactions have been ignored. In fact, the main purpose of this step is to determine an approximate range of neighbors to reduce the time for calculating $P_{jk \to i}(\forall j \neq k \neq i)$. Without taking into account three-body interactions, the expected number of susceptible nodes being infected at $t_m + 1$ can simply be expressed as

$$\tilde{E}_i^{t_m+1} = \sum_{j(j \neq i)} P\left( S_i^{t_m+1} = 1, j \to i \middle| S_i^{t_m} = 0, S_j^{t_m} = 1 \right) * \Psi_j^{t_m} + \varepsilon_i$$
$$= \sum_{j(j \neq i)} P_{j \to i}^0 P_j^i \Psi_j^{t_m} + \varepsilon_i, \tag{19}$$

where the notation $P_{j \to i}^0$ is used to emphasize that node $j$ is only an "approximate" neighbor of node $i$. Assuming that the number $\Psi_i$ of times of node $i$ being infected in each time period obeys the Poisson distribution, we obtain the likelihood function as

$$P\left( \left\{ \Psi_i^{t_m+1} \right\}_{m=1,\cdots,M} \middle| \tilde{\Theta}, \left\{ \Psi_j^{t_m} \right\}_{m=1,\cdots,M; j=1,\cdots,N} \right)$$
$$= \prod_{m(\Psi_i^{t_m} = 0)} \frac{e^{-\tilde{E}_i^{t_m+1}} \left( \tilde{E}_i^{t_m+1} \right)^{\Psi_i^{t_m+1}}}{\Psi_i^{t_m+1}!}, \tag{20}$$

where $\tilde{\Theta}$ denotes the set of variables $P_{j \to i}^0$ and $\varepsilon_i$. Taking the logarithm of Eq. 20, we have

$$L(\tilde{\Theta}) = \sum_{m(\Psi_i^{t_m} = 0)} \left( \Psi_i^{t_m+1} \log \tilde{E}_i^{t_m+1} - \tilde{E}_i^{t_m+1} \right) = \sum_{m(\Psi_i^{t_m} = 0)} \left[ \begin{array}{c} \Psi_i^{t_m+1} \log\left( \sum\limits_{j(j \neq i)} P_{j \to i}^0 P_j^i \Psi_j^{t_m} + \varepsilon_i \right) \\ - \left( \sum\limits_{j(j \neq i)} P_{j \to i}^0 P_j^i \Psi_j^{t_m} + \varepsilon_i \right) \end{array} \right]. \tag{21}$$

Using the EM method to maximize the likelihood function, we obtain the final parameters $\tilde{\Theta}$ as

$$P_{j \to i}^0 = \frac{\sum\limits_{m(\Psi_i^{t_m} = 0)} \left( \Psi_i^{t_m+1} \rho_j^{t_m} \right)}{\sum\limits_{m(\Psi_i^{t_m} = 0)} \left( P_j^i \Psi_j^{t_m} \right)}, \tag{22}$$

$$\varepsilon_i = \frac{\sum\limits_{m(\Psi_i^{t_m} = 0)} \left( \Psi_i^{t_m+1} \rho_{\varepsilon_i}^{t_m} \right)}{\sum\limits_{m(\Psi_i^{t_m} = 0)} (1)}, \tag{23}$$

where

$$\rho_j^{t_m} = \frac{P_{j \to i}^0 P_j^i \Psi_j^{t_m}}{\sum\limits_{j'(j' \neq i)} P_{j' \to i}^0 P_{j'}^i \Psi_{j'}^{t_m} + \varepsilon_i}, \tag{24}$$

$$\rho_{\varepsilon_i}^{t_m} = \frac{\varepsilon_i}{\sum\limits_{j'(j' \neq i)} P_{j' \to i}^0 P_{j'}^i \Psi_{j'}^{t_m} + \varepsilon_i}. \tag{25}$$

With the initial conditions for $P_{j \to i}^0$ and $\varepsilon_i$, the values of $P_{j \to i}^0$ and $\varepsilon_i$ can be obtained by iterating Eqs. 22–25 until convergence is achieved. It is worth noting that $P_{j \to i}^0$ is a probability and we need to determine the "approximate" neighbors of the node under reconstruction. Theoretically, the "approximate" neighbors can be determined by testing whether $P_{j \to i}^0$ is non-zero. However, practically this is not feasible due to noise or deviations from the assumptions. For example, as shown in Fig. 4f, nodes 6 and 14 are not neighbors of node 13 even though $P_{6 \to 13}^0 = 0.0002$ and $P_{14 \to 13}^0 = 0.0006$. To overcome this difficulty, we articulate a truncation method for determining the neighbors of node $i$, as follows.

First, note that the time complexity of the second step can be significantly reduced when fewer neighbors are predicted, but too few predicted neighbors can lead to missing neighbors. On the contrary, too many predicted neighbors would increase the time complexity. A solution is to use a reasonable truncation to determine the "approximate" neighbors of each node. To this end, we re-rank the probability $P_{j \to i}^0 (\forall j \neq i)$ in a descending order and place a threshold $\Delta_i$ in the maximum gap defined as[14]:

$$\Delta_i = \arg\max_l \left[ \frac{P_l'}{P_{l+1}'} \left( P_l' - P_{l+1}' \right) \right]. \tag{26}$$

Next, we use Eq. 26 again to find a new threshold $\bar{\Delta}_i$ which is smaller than $\Delta_i$. Finally, node $j$ is regarded as an "approximate" neighbor of node $i$ if $P_{j \to i}^0 > \bar{\Delta}_i$. The truncation method can ensure the detection of all real neighbors and 2-simplexes.

Once the "approximate" neighbors of node $i$ have been obtained, the time series of these neighbors can be extracted (Fig. 4f, g). The neighbors of node $i$ and its

2-simplexes can be quickly re-predicted based on the second step, i.e., by iterating Eqs. 9–11 and Eqs. 16–18 based on the compressed data matrix. For example, the prediction results for node 13 are shown in Fig. 4h and the values of $P_{j \to i}(\forall j \neq i)$ and $P_{jk \to i}(\forall j \neq k \neq i)$ for each node are presented in Fig. 4i. The actual two- and three-body connections of each node can then be determined based on the results in Fig. 4i. Because the identification of two-body connections has been refined in the second step, we simply assume that node $j$ is a neighbor of node $i$ if $P_{j \to i} > 0$. Following previous work[14,58], we assume that nodes $i$ and $j$ are connected when $P_{j \to i} > 0$ or $P_{i \to j} > 0$.

The case of three-body interactions is more complicated and the solution is sensitive to noise or errors. In fact, using the condition $P_{jk \to i} > 0$ as a criterion to detect $(i, j, k)$ as a 2-simplex can lead to many false positives. Our solution is to re-rank $P_{jk \to i}(\forall j \neq k \neq i)$ in a descending order and obtain a new threshold $\hat{\Delta}_i$ by using Eq. 26 again. As a result, an actual 2-simplex $(i, j, k)$ is formed when $P_{jk \to i} \geq \hat{\Delta}_i$. To remove the conflicts in the prediction, we assume that there exists a 2-simplex $(i, j, k)$ when two of three conditions hold at least, e.g., $P_{jk \to i} \geq \hat{\Delta}_i$, $P_{ik \to j} \geq \hat{\Delta}_j$, and $P_{ij \to k} < \hat{\Delta}_k$, but a three-body cannot form when $P_{jk \to i} \geq \hat{\Delta}_i$, $P_{ik \to j} < \hat{\Delta}_j$, and $P_{ij \to k} < \hat{\Delta}_k$. Implementing the two-step strategy, we can reconstruct the whole 2-simplicial complexes. As shown in Fig. 4a, the 2-simplicial complex has been accurately reconstructed. Overall, the two-step strategy not only greatly reduces the computational time but also significantly improves the reconstruction accuracy (more details in Sec. II in SI).

### Construction of synthetic and real-world 2-simplicial complexes

*Synthetic 2-simplicial complexes*. Here we describe the main steps of constructing synthetic 2-simplicial complexes of size $N$, average degrees of two-body and three-body interactions $k_1$ and $k_2$, respectively.

Random simplicial complex (ERSC): First, a random graph is generated by connecting any two nodes with the probability $p_1$. We then add 2-simplexes between any three nodes with the probability $p_2$, where the formulas of $p_1$ and $p_2$ are[34]:

$$p_1 = \frac{k_1 - 2k_2}{(N-1) - 2k_2},\tag{27}$$

$$p_2 = \frac{2k_2}{(N-1)(N-2)}.\tag{28}$$

A random 2-simplicial complex with the specified average degrees can then be constructed using the probabilities $p_1$ and $p_2$.

Scale-free simplicial complex (SFSC): First, a scale-free network is generated, in which each new node connects $m$ edges to the old nodes with degree preference[59]. We then add 2-simplexes between any three nodes according to probability $p_2$ in Eq. 28. The average degree of 1-simplexes can be calculated as

$$k_1 = 2m + 2k_2\left(1 - \frac{2m}{N}\right).\tag{29}$$

Small-world simplicial complex (SWSC): First, a small-world network[60] is generated from a regular lattice (all the nodes have the same degree $2m$) with rewiring probability $p$. We then add 2-simplexes between any three nodes according to probability $p_2$ in Eq. 28. The average degree of 1-simplexes is given by Eq. 29.

*2-simplicial complexes from real-world data*. In each real-world data set, the face-to-face interactions have been measured with a temporal resolution of 20 s. First, we generate a weighted network according to the data, where a weight represents the number of interactions between a pair of nodes in the whole time window. Second, we remove any link whose weight is less than a given threshold $\zeta$ and set the weights of retained links to one to generate an unweighted network. Finally, we cut the data into multiple segments with a temporal window of 5 min and record all the 2-simplexes. In particular, if three nodes communicate with each other in a short time, they are regarded as constituting a three-body connection. We record the frequencies of the 2-simplexes in each segment. According to the total frequency in all segments, we retain the first 50% of the 2-simplexes with the highest frequencies and count them as the actual 2-simplexes. The visualization of four real-world 2-simplicial complexes is shown in Fig. 2a–d.

### Data availability
The SocioPatterns datasets were downloaded from http://www.sociopatterns.org/datasets/[61]. The source data generated in this study have been deposited on GitHub at: https://github.com/HuanWang2022/reconstruct_simplicial_complex.

### Code availability
The code and datasets are available at: https://github.com/HuanWang2022/reconstruct_simplicial_complex[62].

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

## Acknowledgements

H.W., C.M., H.-S.C. and H.-F.Z. acknowledge supports from the National Natural Science Foundation of China (61973001, 12005001, 11875069), the Natural Science Foundation of Anhui Province (2008085QF299), and the University Synergy Innovation Program of Anhui Province (GXXT-2021-032). Y.-C.L. acknowledges support from the Office of Naval Research under Grant No. N00014-21-1-2323.

## Author contributions

H.W. and C.M. have contributed equally to this work. H.W., C.M., H.-S.C., Y.-C.L., and H.-F.Z. designed the research project, the models and methods. H.W. and C.M. performed the numerical analysis. H.-F.Z. and Y.-C.L. wrote the paper.

## Competing interests

The authors declare no competing interests.
