## [Peer Review File · Nature Communications]

Full reconstruction of simplicial complexes from binary contagion and Ising dataREVIEWER COMMENTS

Reviewer #1 (Remarks to the Author):

The paper "Full reconstruction of simplicial complexes from binary timeseries data" proposes a method to recover the underlying simplicial complex structure (up to 2-simplices, that is, edged and triangles) from the the observation of the (simplicial) contagion dynamics on the node.

In particular, the authors propose an expectation maximization method based on a likelihood function which contains terms describing the probability of a node to be infected at a certain time either by an infectious neighbour, or by a pair of infected neighbours within a 2-simplex, following the simplicial contagion model proposed in ref [33]. They then simulate this dynamical model on various simplicial complexes, both obtained from random ensembles and from data, and show that --given sufficiently long timeseries--, they are able to reconstruct the underlying simplicial complex with good accuracy. Finally, they perturb the original data with progressively growing noise and show that the method is sufficiently robust to noise in the timeseries.

I think the paper is an interesting and timely contribution given the large interest in higher-order representations (hypergraphs, simplicial complexes) for complex systems. Indeed, one of the most important problems right now is how to reconstruct higher-order interactions from time-series data.

The proposed method however is quite limited in its applicability being strongly model-dependent (i.e. on the simplicial contagion model of Ref. [33]), which reduces its overall impact and usability. Overall, I think the paper is well written and sufficiently clear in its exposition of the topic. However, I have a series of comments that I think should be addressed before the paper could be considered for publication in Nature Communications.

Major comments:

- the title is too broad: the authors adopt a specific underlying dynamical model and construct the likelihood function using a contagion dynamics; it is true that they keep the dynamical parameters unknown but still the functional form of the dynamics is mirrored in the likelihood.

I would recommend changing it to something more specific like ".. reconstruction from binary (simplicial) contagion data" or similar.

- this brings me to the next point: how general is this inference scheme for other binary dynamical models, for example what if we consider cascades (e.g. a simplicial version of the cascading processes studied in the SIs of Petri, G., & Barrat, A. (2018). Simplicial activity driven model. Physical review letters, 121(22), 228301.). Would it still work?

- from the methodological and applicative perspectives, it would be interesting to understand how much data is required to reach a certain F1 score. The authors show this in Figure 2 and 3 using simulation time, but I think this is misleading because it obfuscates the dependence on size. In particular, I think the number of interactions/events would be a better measure.

I recommend reproducing those figures showing the results as a function of the number of interactions (potentially infectious events) until that time normalized by the system's size N .

I would also recommend reproducing the reconstruction of simplicial complexes for different sizes, to understand whether there is any effect on F1 given by the size.

- along the same lines, is there a difference in reconstruction accuracy under different epidemic conditions? In the paper it is not clear in what regime(s) (above/below epidemic threshold for edge/triangles) the simulations were performed. Above the triangle-driven transition and below the edge-driven transition, where there is bistability, we likely have a different effect of triangles on the timeseries and thus on the reconstruction, with respect to the regime above the edge-driven transitions and below the triangle-driven one. Does this happen? Whether the answer is yes or no, it is an important thing to know for applications.

- again along the same lines, what about the non-epidemic regime? In this case the timeseries would rapidly go to zero and stay there. However, using repeated outbreaks one could try to estimate the underlying complex as described here. Does it work?

- I do not understand whether the method consistently infers all the subfaces (edges) for the higher-order simplices (triangles, in this case). In fact, until the authors introduce the two-step reconstruction strategy (as a method to improve the data efficiency and computational complexity of the problem) there is no explicit mention of the closure condition required by simplicial complexes to be valid. So, is the two-step strategy required to ensure this, or does the method in itself infer valid complexes, rather than hypergraphs?

- There is also a notable lack of comparison with similar method that extract higher-order interactions. The model the authors use (especially in the long T regimes adopted in the paper) is effectively a stationary model, so the timeseries are amenable to the information-theoretic methods developed by Rosas et al. (Rosas, Fernando E., et al. "Quantifying high-order interdependencies via multivariate extensions of the mutual information." *Physical Review E* 100.3 (2019): 032305.). The latter is not built to create simplicial complexes, but it associates an information-theoretic contribution to the interaction of 3 variables on top of the pairwise marginals. I think it would be important to compare the results of the two reconstructions in order to understand how much the temporal structure outperforms the stationary distribution descriptions on which the Rosas method is based.

Reviewer #2 (Remarks to the Author):

In this manuscript, the authors use SIS contagion time-series data to reconstruct synthetic and empirical simplicial complexes composed of 1- and 2-simplices using an MLE framework. They examine the robustness of their results to stochastic noise introduced into the time-series. Lastly, they leverage the simplicial structure of the data to improve the computational efficiency of their algorithm. For the small datasets that they consider and adequate time-series data, they see remarkable performance.

I found this work to be interesting and relevant to the field. The use of simplicial structure to improve the algorithm was very interesting. The authors comment that they infer both 1- and 2-simplices, not just 2-simplices as in the recent work "Hypergraph reconstruction from network data" by Young and Peixoto.

After reading the manuscript, the methodology and results seem sound, however, there are necessary details missing described below. With these considerations in mind, I recommend that this manuscript be accepted with significant revisions.

Comments:

The authors refer to “high-order” interactions, which implies interactions of large cardinality. However, because the authors only consider 1- and 2-simplex interactions, I would suggest using the standard terminology “higher-order”.

The basic algorithm that the authors describe seems to be applicable to a hypergraph whereas the 2-step method explicitly relies on a simplicial complex. I would recommend making this distinction earlier on in the paper.

Setting $\beta_2 > \beta_1$ seems a bit like an arbitrary choice. What regime are you talking about? How do these values relate to the critical values of β_1 and β_2 ?

Why contagion dynamics? Why not another common opinion model?

How are your results robust to the time step chosen? Also, it wasn't clear that you were using discrete dynamics until the Methods section.

Does increasing the network size make it harder to recover the true structure? How does the time, T , needed to recover the structure depend on the number of nodes and number of 1- and 2-simplices?

You referenced “Method” not “Methods” on page 5.

Figure 5 is very nice.

I'm confused as to why the methodology of reconstruction is after the discussion and not prior to your results. I would suggest moving your methodology before the results, because your methodology seems to be the main contribution of this work and it would be good to motivate Figures 2, 3, and 4.

In Figure 4, where you show the F1 score with respect to the flip probability, can you explain your results a bit more? For example, why are the 2-simplices for random networks more sensitive (looked like 2x more sensitive) to perturbations than for the empirical datasets?

Can one infer based on the time-series data that the assumed model is incorrect in contrast to poor recovery?

In the Methods section, can you move Figure 5 to the main text? It was hard scrolling back and forth.

How do you expect that changing β_1 and/or β_2 will affect your results?

Sorry if I missed this, but what is the variable C where you describe the complexity of your 2-step method?

Did you compare the 1-step method to the 2-step method? I saw neither a time complexity comparison nor an accuracy comparison based on the mean degrees.

In several of the figures, it would be helpful if you used more grayscale-friendly colors.

I would suggest that the authors make their code publically available if possible so that it is easier to verify their results.

Point-by-point response to referee comments

Referee 1

General comment: *“The paper “Full reconstruction of simplicial complexes from binary time-series data” proposes a method to recover the underlying simplicial complex structure (up to 2-simplices, that is, edged and triangles) from the observation of the (simplicial) contagion dynamics on the node. In particular, the authors propose an expectation maximization method based on a likelihood function which contains terms describing the probability of a node to be infected at a certain time either by an infectious neighbour, or by a pair of infected neighbours within a 2-simplex, following the simplicial contagion model proposed in ref [33]. They then simulate this dynamical model on various simplicial complexes, both obtained from random ensembles and from data, and show that –given sufficiently long timeseries–, they are able to reconstruct the underlying simplicial complex with good accuracy. Finally, they perturb the original data with progressively growing noise and show that the method is sufficiently robust to noise in the timeseries.*

I think the paper is an interesting and timely contribution given the large interest in higher-order representations (hypergraphs, simplicial complexes) for complex systems. Indeed, one of the most important problems right now is how to reconstruct higher-order interactions from time-series data. The proposed method however is quite limited in its applicability being strongly model-dependent (i.e. on the simplicial contagion model of Ref. [33]), which reduces its overall impact and usability. Overall, I think the paper is well written and sufficiently clear in its exposition of the topic. However, I have a series of comments that I think should be addressed before the paper could be considered for publication in Nature Communications.”

Response: We thank the referee for evaluating our work positively. We have carried out new analysis and computations and made appropriate revisions to fully to address all his/her comments.

A major change is that we have introduced a simplicial Ising dynamics (Sec. IIA in the main text and Sec. III in Supplementary Information) to demonstrate that our reconstruction framework is applicable to binary dynamical processes beyond the simplicial contagion model.

We note that, even though different types of dynamical processes on simplicial complexes have been studied, not all of them are suitable for the task of network reconstruction. Our methodology is essentially a statistical inference framework, so it can be naturally applied to discrete-state dynamics, insofar as the dynamics directly reflect the synergistic reinforcement effects from a 2-simplex (i.e., three-body interactions).

Comment 1: *“The title is too broad: the authors adopt a specific underlying dynamical model and construct the likelihood function using a contagion dynamics; it is true that they keep the dynamical parameters unknown but still the functional form of the dynamics is mirrored in the likelihood. I would recommend changing it to something more specific like “... reconstruction from binary (simplicial) contagion data” or similar.”*

Response. In the revised manuscript, the examples demonstrating the working of our methodology are social contagion and Ising type of binary dynamics (Sec. IIA in the main text and Sec. III in Supplementary Information). We have followed the referee suggestion by changing the title to a more specific one: “Full reconstruction of simplicial complexes from binary contagion and Ising data”.

Comment 2: *“This brings me to the next point: how general is this inference scheme for other binary dynamical models, for example what if we consider cascades (e.g. a simplicial version of the cascading processes studied in the SIS of Petri, G., & Barrat, A. (2018). Simplicial activity driven model. Physical review letters, 121(22), 228301.). Would it still work?”*

Response: We have introduced a simplicial Ising model to verify that our method is applicable to other binary dynamical processes in addition to contagion dynamics. The detailed derivations of the simplicial Ising system and the results are presented in Sec. III in Supplementary Information.

After extensive testing, we have found that it is difficult to apply our statistical based reconstruction framework to time-varying, simplicial activity driven networks. The main reason is that even reconstructing conventional time-varying networks is already extremely difficult because the structure of the network is not fixed but changes with time. The corresponding task of reconstructing simplicial complexes can be much harder - we are not aware of any work on this subject. The propagation dynamics proposed in PRL, 121(22), 228301 (2018) (cited as Ref. [38] in the revised manuscript) is not suitable for reconstructing higher-order interactions in simplicial complexes because the synergistic reinforcement effect from 2-simplex is not *directly* reflected in the underlying propagation process.

Comment 3: *“From the methodological and applicative perspectives, it would be interesting to understand how much data is required to reach a certain F1 score. The authors show this in Figure 2 and 3 using simulation time, but I think this is misleading because it obfuscates the dependence on size. In particular, I think the number of interactions/events would be a better measure. I recommend reproducing those figures showing the results as a function of the number of interactions (potentially infectious events) until that time normalized by the system’s size N . I would also recommend reproducing the reconstruction of simplicial complexes for different sizes, to understand whether there is any effect on F1 given by the size.”*

Response: Following referee’s suggestion, we present here F1 score as a function of the number of events for scale-free simplicial complex (SFSC) (Response Fig. 1) and four real-world 2-simplicial complexes (Response Fig. 2), where the number of events is calculated on each node. For example, 100 events means that each node has 100 switches between susceptible and infected states. It can be seen that the reconstruction accuracy increases with the number of events. In addition, the reconstruction accuracies of two-body and three-body connections decrease with k_1 (i.e., average degree of 1-simplex), but the value of k_2 (i.e., average degree of 2-simplex) affects only the accuracy of three-body connections and has little effect on the accuracy of reconstructing two-body connections. While a larger simplicial complex requires more events to reach certain F1 score, with sufficient data a high reconstruction accuracy can still be achieved.

We have added a new section (Sec. IV) in SI to present these results. The conclusion is that the results based on the number of events are essentially the same as those in terms of the simulation time T .

Comment 4: *“Along the same lines, is there a difference in reconstruction accuracy under different epidemic conditions? In the paper it is not clear in what regime(s) (above/below epidemic threshold for edge/triangles) the simulations were performed. Above the triangle-driven transition and below the edge-driven transition, where there is bistability, we likely have a different effect of triangles on the timeseries and thus on the reconstruction, with respect to the regime above the edge-driven transitions and below the triangle-driven one. Does this happen? Whether the answer is yes or no, it is an important thing to know for applications.”*

Response Figure 1: F1 score as a function of the number of events for SFSC for (a,d) $N = 100$, $\alpha = 0.6$, $\omega = 2$, (b,e) $N = 200$, $\alpha = 0.7$, $\omega = 2.2$, and (c,f) $N = 300$, $\alpha = 0.75$, $\omega = 2.3$. In each panel, squares, diamonds and circles demonstrate the performance of reconstructing two-body connections, and triangles with different orientations depict the performance of reconstructing three-body connections. The average degrees are distinguished by colors. Other parameter values are $\rho_0 = 0.2$ and $\mu = 1$. Each data point is the result of averaging over five realizations.

Response. Yes, there is a difference in the reconstruction accuracy under different epidemic conditions. In our study, the parameter values for generating the data are selected near the epidemic threshold, i.e., near the triangle-driven and edge-driven transitions. To assess the effects of choosing parameter values in other regimes on the reconstruction results, we focus on two quantities: the rescaled edge infectivity α and the rescaled triangular infectivity ω , and study their impact on the reconstruction accuracy of two-body and three-body connections (Sec. I in Supplementary Information).

Some representative results are as follows. Response Fig. 3 shows the average fraction ρ^* of infected nodes in the stationary state versus α for different values of ω . It can be seen that, as ω increases, the nature of the phase transition in the underlying social contagion dynamics changes from continuous to discontinuous, which is consistent with the results in Iacopini et al., Nat. Commun. 10, 1-9 (2019).

We then choose three different values of α (0.5, 0.8 and 1.5), which are below, near and above the edge epidemic threshold α_c , respectively, and investigate the effect of different values of ω on the reconstruction accuracy. Concretely, we choose $\omega = 0.8, 2.4$ and 4.0 , which are below, near and above the triangular epidemic threshold ω_c , respectively. Response Fig. 4(a) reveals that, for $\alpha = 0.5$, the reconstruction accuracies of two-body and three-body connections are not high, because it is difficult for social contagion in simplicial complexes to propagate when the rescaled edge infectivity α is small. Increasing the value of ω can lead to an improvement in the reconstruction accuracy. For example, as shown in Response Fig. 4(b),

Response Figure 2: F1 score as a function of the number of events for four real-world 2-simplicial complexes: (a) Hypertext2009, (b) Thiers12, (c) InVS15, (d) LyonSchool. In each panel, the blue squares and red circles demonstrate the performance of reconstructing two-body and three-body connections, respectively. Parameter values are $\alpha = 0.3$, $\omega = 1$, $\rho_0 = 0.2$ and $\mu = 1$. The “ N ” appeared in the abscissa denotes the size of the specific network. Each data point is the result of averaging over five realizations.

for $\alpha = 0.8$ and reasonably long time series, the reconstruction accuracies of two-body and three-body connections for $\omega = 2.4$ are the highest, as a small value of ω means an insignificant synergistic reinforcement effect from the three-body connections but a large value of ω will weaken the interactions from the two-body connections, especially when the value of α is near the edge threshold: both effects lead to difficulties in reconstructing two-body or three-body connections. Response Fig. 4(c) shows, for $\alpha = 1.5$, the reconstruction accuracies of two-body and three-body connections for $\omega = 2.4$ are generally higher than those in the other two cases, because a small value of ω (e.g., $\omega = 0.8$) is not able to generate a strong synergistic reinforcement effect from the three-body connections while a large value of ω (e.g., $\omega = 4$) will cause most nodes to be infected. Taken together, the highest possible reconstruction accuracies are achieved when the values of α and ω are near their respective epidemic thresholds.

Response Figure 3: The average fraction ρ^* of infected nodes in the stationary state as a function of the rescaled edge infectivity (α) for SFSC. The curves correspond to different values of the rescaled triangular infectivity ω . Other parameter values are $\rho_0 = 0.2$ and $\mu = 1$, and 50 statistical realizations are used to generate each data point.

Response Figure 4: Reconstruction performance for different values of α and ω . Shown is F1 score as a function of the time series length T for SFSC for (a) $\alpha = 0.5$, (b) $\alpha = 0.8$, and (c) $\alpha = 1.5$. In each panel, squares, diamonds and circles denote the accuracy of reconstructing two-body connections, while triangles with different orientations denote the accuracy of reconstructing three-body connections. The results from different values of ω are distinguished by colors. Other parameter values are $\rho_0 = 0.2$ and $\mu = 1$, and five realizations are used to generate the results.

Comment 5: “Again along the same lines, what about the non-epidemic regime? In this case the time-series would rapidly go to zero and stay there. However, using repeated outbreaks one could try to estimate the underlying complex as described here. Does it work?”

Response. In the non-epidemic regime, the underlying simplicial complex structure cannot be reconstructed efficiently even using repeated outbreaks. The reason is that, when the infection rate is too small, few useful data can be obtained even from very long time series, i.e., the number of transitions from the susceptible to the infected state is very low. In this case, to reconstruct the simplicial complex structure is not possible, as shown in Response Fig. 5, where two parameter settings are used: (1) $\alpha = 0.5$ and $\omega = 0.4$, and (2) $\alpha = 0.5$ and $\omega = 0.8$. In both cases, the F1 score is low, regardless of the nature of the interactions, i.e., two-body or three-body.

Response Figure 5: Reconstruction performance for smaller values of α and ω . Shown is F1 score versus the time-series length T for SFSC. Squares and circles correspond to the case of reconstructing two-body connections while triangles with different orientations are for reconstructing three-body connections. The cases with different values of ω are distinguished by colors. The simplicial complex size is $N = 200$. Other parameter values are $k_1 = 14$, $k_2 = 4$, $\rho_0 = 0.2$ and $\mu = 1$, and five realizations are used to generate each data point.

Comment 6: “I do not understand whether the method consistently infers all the subfaces (edges) for the higher-order simplices (triangles, in this case). In fact, until the authors introduce the two-step reconstruction strategy (as a method to improve the data efficiency and computational complexity of the problem) there is no explicit mention of the closure condition required by simplicial complexes to be valid. So, is the two-step strategy required to ensure this, or does the method in itself infers valid complexes, rather than hypergraphs?”

Response. Our two-step method was not designed for the extremely challenging task of consistently inferring all the subfaces for *arbitrary* higher-order simplices. In fact, our method requires the closure condition of simplicial complexes: it is necessary to know in advance that the network under reconstruction is a

2-simplicial complex. Given this premise, the two-step strategy infers first the two-body and then the three-body interactions (i.e., 2-simplex) from the inferred two-body interactions. While the two-step method is efficient to reconstruct 2-simplicial complexes, at the present it cannot be used to reconstruct the hypergraphs because its second step is to find the triangles from the neighbors (i.e., edges). In Sec. IVA3 of the revised manuscript, we have added the following explanation to highlight the scope of application of our two-step strategy:

- Our two-step method was not designed for the general and extremely challenging task of consistently inferring all the subfaces for *arbitrary* higher-order simplices. In fact, our method requires the closure condition of simplicial complexes: it is necessary to know in advance that the network under reconstruction is a 2-simplicial complex. Given this premise, the two-step strategy infers first the two-body and then the three-body interactions (i.e., 2-simplex) from the inferred two-body interactions. While the two-step method is efficient to reconstruct 2-simplicial complexes, at the present it cannot be used to reconstruct the hypergraphs because its second step is to find the triangles from the neighbors (i.e., edges).

Comment 7: *“There is also a notable lack of comparison with similar method that extract higher-order interactions. The model the authors use (especially in the long T regimes adopted in the paper) is effectly a stationary model, so the timeseries are amenable to the information-theoretic methods developed by Rosas et al. (Rosas, Fernando E., et al. “Quantifying high-order interdependencies via multivariate extensions of the mutual information.” Physical Review E 100.3 (2019): 032305.). The latter is not built to create simplicial complexes, but it associates an information-theoretic contribution to the interaction of 3 variables on top of the pairwise marginals. I think it would be important to compare the results of the two reconstructions in order to understand how much the temporal structure outperforms the stationary distribution descriptions on which the Rosas method is based.”*

Response. In the recent work pointed out by the referee [PRE 100, 032305 (2019)], an important information metric named O-information (Ω) was introduced, which can be used to characterize synergy- and redundancy-dominated systems and quantify higher-order interdependencies. It can be seen from Lemma 1 in the paper that O-information can only capture the interactions that go beyond the pairwise relationships, so it is not suitable for describing the interaction of two variables, i.e., it cannot be used to reconstruct the pairwise relationships.

More Specifically, for a system of three discrete variables, the O-information is defined as

$$\Omega(X^n) = H(X^n) + \sum_{j=1}^n [H(X_j) - H(X_{-j}^n)],$$

where

$$\begin{aligned} H(X^n) &= - \sum_{X^n} P_{X^n}(X^n) \log P_{X^n}(X^n), \\ X^n &= (X_1, \dots, X_n), \\ X_{-j}^n &= (X_1, \dots, X_{j-1}, X_{j+1}, \dots, X_n), \end{aligned}$$

for $n = 3$. According to this metric, the system is redundancy dominated if $\Omega(X^3) > 0$; otherwise ($\Omega(X^3) < 0$), it is synergy dominated. To compare our method with this O-information based method,

we have calculated the O-information values between any three variables. If it is negative, there is an interaction among the three variables (three-body connection), otherwise the connection does not exist. The results are shown in Response Fig. 6, where the O-information values of any three points are displayed with the blue and red dots denoting the existent and nonexistent three-body connections, respectively. It can be seen that the O-information values associated with most of the existent three-body interactions are indeed negative, but the O-information values of many nonexistent three-body interactions are also negative, making it impossible to distinguish the two cases and to ascertain the existent three-body interactions.

Overall, the O-information method proposed in [PRE 100, 032305 (2019)] relies on strong correlations among the time series of the dynamical variables for predicting the synergy structure. In our case, the dynamical time-series data are obtained by alternating updated iterations, that is, the state transition of each node is determined by the states of neighbors at the previous time, leading to only weak correlations among different nodal pairs at any given time. As a result, the O-information method is not suitable for predicting higher-order structures from dynamical time-series data.

Response Figure 6: Performance of O-information based method. Shown are the O-information values of any three points $\Omega(X^3)$ on ERSC. Each row corresponds to the O-information of one node. The blue and red dots denote the existent and nonexistent three-body connections, respectively. The parameter values are $N = 100$, $T = 10000$, $k_1 = 7$, $k_2 = 2$, $\alpha = 0.6$, $\omega = 2.1$, $\rho_0 = 0.2$ and $\mu = 1$. For each node, the number of all possible three-body interactions is about N^2 . Because the number of existent three-body connections is very low, only 0.2% of the red dots are shown in each row for better visualization.

We have added a new section (Sec. V) in SI to discuss the O-information measure.

Referee 2

General comment: *“In this manuscript, the authors use SIS contagion time-series data to reconstruct synthetic and empirical simplicial complexes composed of 1- and 2-simplices using an MLE framework. They examine the robustness of their results to stochastic noise introduced into the time-series. Lastly, they leverage the simplicial structure of the data to improve the computational efficiency of their algorithm. For the small datasets that they consider and adequate time-series data, they see remarkable performance.*

I found this work to be interesting and relevant to the field. The use of simplicial structure to improve the algorithm was very interesting. The authors comment that they infer both 1- and 2-simplices, not just 2-simplices as in the recent work “Hypergraph reconstruction from network data” by Young and Peixoto.

After reading the manuscript, the methodology and results seem sound, however, there are necessary details missing described below. With these considerations in mind, I recommend that this manuscript be accepted with significant revisions.”

Response: We appreciate the referee’s insightful and positive comments. We have carried out new analysis and computations to fully address all the comments.

Comment 1: *“The authors refer to “high-order” interactions, which implies interactions of large cardinality. However, because the authors only consider 1- and 2-simplex interactions, I would suggest using the standard terminology ‘higher-order’.”*

Response: Agreed. We have replaced the term “high-order” by “higher-order” throughout the text.

Comment 2: *“The basic algorithm that the authors describe seems to be applicable to a hypergraph whereas the 2-step method explicitly relies on a simplicial complex. I would recommend making this distinction earlier on in the paper.”*

Response: The referee is completely correct that the two-step method cannot be applied to hypergraphs because the method requires the a priori condition of 2-simplicial complexes, i.e., it is necessary to assume that network under reconstruction is a 2-simplicial complexes. Under this assumption, our two-step method first infers the two-body interactions and predicts the three-body interactions (i.e., 2-simplex) from the inferred two-body interactions. While the two-step method is efficient to reconstruct 2-simplicial complexes, at the present it cannot reconstruct the hypergraphs because its second step is to find the triangles from the neighbors (i.e., edges). In Sec. IVA3 of the revised manuscript, we have added the following explanation to highlight the scope of application of our two-step strategy:

- Our two-step method was not designed for the general and extremely challenging task of consistently inferring all the subfaces for *arbitrary* higher-order simplices. In fact, our method requires the closure condition of simplicial complexes: it is necessary to know in advance that the network under reconstruction is a 2-simplicial complex. Given this knowledge, the two-step strategy infers first the two-body and then the three-body interactions (i.e., 2-simplex) from the inferred two-body interactions. While the two-step method is efficient to reconstruct 2-simplicial complexes, at the present it cannot reconstruct the hypergraphs because its second step is to find the triangles from the neighbors (i.e., edges).

Comment 3: “Setting $\beta_2 > \beta_1$ seems a bit like an arbitrary choice. What regime are you talking about? How do these values relate to the critical values of β_1 and β_2 ?”

Response: This Comment is essentially the same as Comment 4 of the first referee. Please see our corresponding Response, Response Figs. 3 and 4, and the detailed explanations above.

Comment 4: “Why contagion dynamics? Why not another common opinion model?”

Response. To demonstrate the general applicability of our method, in the revised paper we have introduced a simplicial Ising dynamics (Supplementary Information III). In the Ising model, each node can have two distinct states: spin-up and spin-down, which can be viewed as an alternative model of opinion dynamics. For conventional networks with two-body interactions only, previous works have established the equivalence between the Ising and opinion dynamics, e.g,

- Stauffer D. Social applications of two-dimensional Ising models. *Ame. J. Phys.*, 2008, 76(4): 470-473.
- Grabowski A, Kosiński R A. Ising-based model of opinion formation in a complex network of interpersonal interactions. *Physica A*, 2006, 361(2): 651-664.
- Galam S, Martins A C R. Two-dimensional Ising transition through a technique from two-state opinion-dynamics models. *Phys. Rev. E*, 2015, 91(1): 012108.

Comment 5: “How are your results robust to the time step chosen? Also, it wasn’t clear that you were using discrete dynamics until the Methods section.”

Response. We have systematically studied the impact of the length of the time series on the reconstruction performance, as follows.

The reconstruction performance on synthetic and real-world 2-simplicial complexes versus the length T of the time series has been studied for different noise level (the flip ratio) f : $f = 0$ (without noise), 0.1, 0.2, and 0.3, with results shown in Response Figs. 7 and 8, respectively. These results indicate that, for short time series, the reconstruction performance is sensitive to noise. For example, when the value of f changes from 0 to 0.1, there is a sizable reduction in the F1 score. Regardless of the noise level, increasing the length of time series can always improve the reconstruction performance for both two-body and three-body interactions.

We have added a new section (Sec. VI) in SI to discuss the effect of the time-series length on the reconstruction.

In the Abstract of the revised manuscript, we have emphasized that, because of its statistical-inference nature, our method is particularly suited for reconstructing 2-simplicial complexes from data generated by discrete dynamics (i.e., simplicial contagion and Ising dynamics).

Comment 6: “Does increasing the network size make it harder to recover the true structure? How does the time, T , needed to recover the structure depend on the number of nodes and number of 1- and 2-simplices?”

Response Figure 7: Reconstruction performance under noise for synthetic 2-simplicial complexes. Shown is F1 score for different values of the flip ratio f as a function of the time-series length T for three synthetic 2-simplicial complexes: random simplicial complex (ERSC - left column), scale-free simplicial complex (SFSC - middle column), and small-world simplicial complex (SWSC - right column): (a-c) reconstructing two-body connections and (d-f) reconstructing three-body interactions. The results for different values of f are distinguished by symbols and colors. All simplicial complexes have the same size $N = 200$. Other parameter values are $k_1 = 12$, $k_2 = 4$, $\alpha = 0.8$, $\omega = 2.4$, $\rho_0 = 0.2$, and $\mu = 1$. Each data point is the result of averaging over five realizations.

Response: Yes, as demonstrated in Response Fig. 9, increasing the network size requires longer time steps for a 2-simplicial complex to be reconstructed.

Response Figs. 10(a)-10(c) show, for a fixed value of k_2 (the number of 2-simplex being $k_2 * N/3$), as the value of k_1 (the number of 1-simplex being $k_1 * N/2$) increases, it becomes more difficult to infer correctly the two-body connections. Similarly, for a fixed value of k_1 , the impact of varying k_2 on the reconstruction has been studied, as shown in Response Figs. 10(d)-10(f). It can be seen that the value of k_2 affects only the reconstruction accuracy of three-body connections and has little effect on the accuracy of reconstructing two-body connections. (Please note that Response Fig. 10 is in fact Fig. 2 in the main text associated with the study of the impact of the number of 1- and 2-simplices on the reconstruction accuracy.)

Comment 7: “You referenced “Method” not “Methods” on page 5.”

Response. It has been fixed.

Response Figure 8: Reconstruction performance under noise for real-world 2-simplicial complexes. Shown is F1 score for different values of the flip ratio f as a function of the time-series length T for (a,d) Thiers12, (b,e) InVS15, and (c,f) LyonSchool: (a-c) reconstructing two-body connections and (d-f) reconstructing three-body interactions. Parameter values are $\alpha = 0.3$, $\omega = 1$, $\rho_0 = 0.2$, and $\mu = 1$. Each data point is the result of averaging over five realizations.

Comment 8: “Figure 5 is very nice. I’m confused as to why the methodology of reconstruction is after the discussion and not prior to your results. I would suggest moving your methodology before the results, because your methodology seems to be the main contribution of this work and it would be good to motivate Figures 2, 3, and 4.”

Response: *Nature Communications* requires that the Methods Section be placed after the Results and Discussion Sections. To facilitate reading, we have moved Fig. 5 to the beginning of the Methods Section, as this figure illustrates the basic idea and principle behind our reconstruction method.

Comment 9: “In Figure 4, where you show the F1 score with respect to the flip probability, can you explain your results a bit more? For example, why are the 2-simplices for random networks more sensitive (looked like 2x more sensitive) to perturbations than for the empirical datasets?”

Response: Because the numbers of 1- and 2-simplicies in empirical networks are very low (as illustrated in Fig. 3 in the main text), it is difficult for a social contagion to propagate on these networks. As a result, the number of infected nodes is low, which impacts negatively on network reconstruction. We thus set a long time interval ($T = 20000$) for empirical networks. As shown in Response Fig. 11, compared with the case of $T = 20000$, the flipping ratio f has a more significant effect on the reconstruction accuracy for

Response Figure 9: Reconstructing synthetic 2-simplicial complexes of varying sizes. Shown is F1 score as a function of the time-series length T for reconstructing three synthetic 2-simplicial complexes: (a) ERSC, (b) SFSC, and (c) SWSC. In each panel, squares, diamonds, circles and hexagons denote the case of reconstructing two-body connections, while triangles with different orientations correspond to reconstructing three-body connections. The sizes of the simplicial complexes are distinguished by colors. For $N = 100, 200, 300,$ and 500 , the parameter values are $(k_1, k_2, \alpha, \omega) = (7, 2, 0.6, 2.1), (14, 4, 0.8, 2.4), (21, 6, 0.8, 2.4),$ and $(35, 10, 0.8, 2.4)$, respectively. Other parameter values are $\rho_0 = 0.2$ and $\mu = 1$. The results are averaged over five realizations.

$T = 10000$.

Due to the difficulty of weak spreading in empirical networks, the number of infected state (i.e., “1” in data matrix S) in empirical networks is much smaller than that in random networks. Take $T = 10000$ as an example. Assume there are 1000 ones in an empirical network and 10000 ones in the random network. With the flipping ratio $f = 10\%$, 100 ones and 100 zeros are flipped, so the fraction of perturbation in the empirical network is $(100 + 100)/(T * N)$. However, for the random network, 1000 ones and 1000 zeros are flipped, so the fraction of perturbation is $(1000 + 1000)/(T * N)$. As a result, even with the same flipping ratio f , the number of transitions in the random network is much larger than that in the empirical network. This explains why reconstructing 2-simplices for random networks is more sensitive to noise than for empirical networks.

Comment 10: “Can one infer based on the time-series data that the assumed model is incorrect in contrast to poor recovery?”

Response. Yes, the reconstruction accuracy is not very good when the recovery is poor. The number of transitions from the susceptible to the infected state should be large because our statistical inference framework relies on the probability of being infected. If the recovery rate μ is small, the infected nodes cannot return to the susceptible state, leading to an unacceptably small number of transition events. Response Fig. 12 demonstrates that the reconstruction accuracy is low for $\mu = 0.1$ but is high for $\mu = 1.0$.

Comment 11: “In the Methods section, can you move Figure 5 to the main text? It was hard scrolling back and forth.”

Response. As explained in our Response to Comment 8, *Nature Communications* requires that the Methods Section be placed after the Results and Discussion sections. We have moved Fig. 5 to the beginning of

Response Figure 10: Reconstruction performance for synthetic 2-simplicial complexes. Shown is F1 score as a function of the length T of the observational binary time series for three synthetic 2-simplicial complexes: random simplicial complex (ERSC - left column), scale-free simplicial complex (SFSC - middle column), and small-world simplicial complex (SWSC - right column). In each panel, squares, diamonds and circles denote the performance of reconstructing two-body connections, triangles with different directions denote the performance of reconstructing three-body connections, and the average degrees are distinguished by colors. All simplicial complexes have the same size $N = 200$. Other parameter values are $\alpha = 0.8$, $\omega = 2.4$, $\rho_0 = 0.2$, and $\mu = 1$. The results are averaged over five realizations.

the Methods section.

Comment 12: “How do you expect that changing β_1 and/or β_2 will affect your results?”

Response. Different values of β_1 and β_2 can have significantly different impacts on the reconstruction results. For large values of β_1 or β_2 , many nodes are infected, making it difficult to judge which infected nodes have spread the infection to the node under reconstruction. On the contrary, for small values of β_1 and β_2 , it is difficult for a susceptible node to be infected by other nodes, leading to a lack of the useful data for the reconstruction task. It is then necessary to select the values of β_1 and β_2 properly to achieve acceptable reconstruction accuracy. In our Response to referee’s Comment 3, we have systematically investigated the different parameter settings (e.g., in terms of the rescaled infectivities α and ω) and their effects on the reconstruction accuracy. (Some details can be found in Sec. I in SI).

Comment 13: “Sorry if I missed this, but what is the variable C where you describe the complexity of your 2-step method?”

Response Figure 11: Effect of random flipping on reconstruction performance for real-world simplicial complexes. Shown is the F1 score for (a) Thiers12, (b) InVS15, (c) LyonSchool. In each panel, squares and circles indicate the performance of reconstructing two-body connections while triangles with different orientations display the performance of reconstructing three-body connections. Different values of the time-series length T are distinguished by colors. The parameter values are $\alpha = 0.3$, $\omega = 1$, $\rho_0 = 0.2$, and $\mu = 1$. Each data point is the result of averaging over ten realizations.

Response Figure 12: Effect of poor recovery on reconstruction performance. Shown is F1 score as a function of the time-series length T for SFSC. Squares and circles indicate the performance of reconstructing two-body connections while triangles with different orientations depict the performance of reconstructing three-body connections. The different values of the recovery rate μ are distinguished by colors. The simplicial complex size is $N = 200$. Other parameter values are $k_1 = 14$, $k_2 = 4$, $\alpha = 0.8$, $\omega = 2.4$ and $\rho_0 = 0.2$. The results are averaged over five realizations.

Response. In the revised manuscript, we have changed C_{N-1}^2 to $\binom{N-1}{2}$, which characterizes the computational complexity of the one-step reconstruction method.

Comment 14: “Did you compare the 1-step method to the 2-step method? I saw neither a time complexity comparison nor an accuracy comparison based on the average degrees.”

Response. In order to compare the accuracy and time complexity of one-step and two-step methods, we have used scale-free simplicial complex (SFSC) with different sizes and average degrees. The results are shown in Response Figs. 13 and 14, as well as in Sec. II in SI). In particular, Response Fig. 13 shows that the accuracy of the two-step method is higher than that of the one-step method for both two-body and three-body reconstruction. The superiority of two-step method is more evident when the network size is larger. Response Fig. 14 demonstrates the running time of the two methods, which are implemented in MATLAB2016a and run on a Linux machine with 2.60-GHz Intel processor, 28 CPU cores, and 192-GB RAM. It can be seen that the required computational time of the two-step method is more than one order of magnitude lower than that required of the one-step method.

Response Figure 13: Comparison of the reconstruction performance between one-step and two-step methods in SFSC with different sizes and degrees. Shown is F1 score as a function of the time-series length T . Squares and circles denote the performance of reconstructing two-body connections while triangles with different orientations are for reconstructing three-body connections. The results from the two methods are distinguished by colors. The parameter values in each simplicial complex are (a) $N = 50, k_1=2, k_2=1, \alpha=0.2, \omega=1$, (b) $N=50, k_1=4, k_2=1, \alpha=0.4, \omega=1$, (c) $N= 100, k_1=8, k_2=2, \alpha=0.6, \omega=1.55$, and (d) $N= 100, k_1=8, k_2=3, \alpha=0.8, \omega=1.6$. Other parameter values are $\rho_0 = 0.2$ and $\mu = 1$. The results are averaged over five realizations.

Response Figure 14: Comparison of the required computational time of the one-step method and two-step method on SFSC with different sizes and degrees. The parameter values are the same as those in Response Fig. 13. The running time is calculated based on one realization.

Comment 15: “In several of the figures, it would be helpful if you used more grayscale-friendly colors.”

Response. In the revised manuscript, some figures have been modified in a grayscale-friendly way. For example, in the previous version, squares and circles denote the performance of reconstructing two-body and three-body connections, respectively. In the revised version, we have used more symbols to represent the results for different parameter settings.

Comment 16: “I would suggest that the authors make their code publically available if possible so that it is easier to verify their results.”

Response. Yes, we have uploaded the codes on GitHub, please see:

https://github.com/HuanWang2022/reconstruct_simplicial_complex.

REVIEWERS' COMMENTS

Reviewer #1 (Remarks to the Author):

I thank the authors for their accurate and thoughtful reply.
I recommend the paper for publication.

Reviewer #2 (Remarks to the Author):

The authors addressed my concerns and suggestions in a satisfactory way. They explored the parameter space more thoroughly, more clearly explained their methodology, studied an additional dynamical system, and made their code publicly accessible. The quality of the manuscript is greatly improved.

A small point is that in response figure 3 (figure S1), for $\omega=2.4,3.2,4$, there should be two stable fixed points and a corresponding hysteresis loop (as in Iacopini et al. 2019 Figs. 2 and 3a). In addition, the authors consider $\rho_0 = 0.2$ and although the manuscript mentions the effect of β_2 and β_3 , a brief mention in the Supplement of the effect of ρ_0 when there is bistable behavior would be helpful.

I recommend publication of this manuscript with these minor changes.

Point-by-point response to referee comments

Referee 1

General comment: *“I thank the authors for their accurate and thoughtful reply. I recommend the paper for publication.”*

Response: We thank the referee for recommending our paper for publication.

Referee 2

General comment: *“The authors addressed my concerns and suggestions in a satisfactory way. They explored the parameter space more thoroughly, more clearly explained their methodology, studied an additional dynamical system, and made their code publicly accessible. The quality of the manuscript is greatly improved.*

...

I recommend publication of this manuscript with these minor changes.”

Response: We thank the referee for his/her positive evaluation and for raising two new minor comments. We have implemented the requested minor changes in the revised manuscript.

Comment 1: *“A small point is that in response figure 3 (figure S1), for $\omega = 2.4, 3.2, 4$, there should be two stable fixed points and a corresponding hysteresis loop (as in Iacopini et al. 2019 Figs. 2 and 3a).”*

Response: Agreed. As shown in the panel (b) of Response Fig. 1 (of this Response), for $\omega = 2.4, 3.2, 4$, there is indeed a bistable region in which the healthy and endemic states co-exist and a hysteresis loop emerges, whose area increases with the value of ω .

This result has been incorporated into the Supporting Information (new Fig. S1 and the accompanying explanations).

Comment 2: *“In addition, the authors consider $\rho_0 = 0.2$ and although the manuscript mentions the effect of β_2 and β_3 , a brief mention in the Supplement of the effect of ρ_0 when there is bistable behavior would be helpful.”*

Response: Thanks for the insightful suggestions. As shown in the panel (b) of Response Fig. 1, the hysteresis loop in each subfigure is indicative of the existence of the bistable behavior. Following referee’s suggestion, we have added the following description of the effect of ρ_0 in Supplementary Information (in the third paragraph on page 2):

- It can be seen that, as ω increases, the nature of the phase transition in the underlying social contagion dynamics changes from continuous to discontinuous. Further, different initial values of the density ρ_0 of the infected nodes can affect the steady-state infection density ρ^* associated with the healthy and endemic states in the bistable region.

The caption of Fig. S1 has been revised accordingly to include a description of the effect of ρ_0 .

Response Figure 1: Average fraction ρ^* of infected nodes in the stationary state as a function of the rescaled edge infectivity α for SFSC. (a) The resulting ρ^* -vs- α curves for different values of the rescaled triangular infectivity ω for $\rho_0 = 0.2$ and $\mu = 1$. As ω increases, the nature of the phase transition in the underlying social contagion dynamics changes from continuous to discontinuous. (b) The effect of the initial density of the infected nodes on the ρ^* -vs- α curve for three values ω (three subpanels). In each subpanel, the ρ^* -vs- α curves for two values of ρ_0 are shown: $\rho_0 = 0.02$ (red circles) and $\rho_0 = 0.8$ (blue squares). Different initial values of the density ρ_0 of the infected nodes can affect the steady-state infection density ρ^* associated with the healthy and endemic states in the bistable region. For all the curves in (a) and (b), each data point is the result of averaging over 50 statistical realizations.